# A 1 km soil moisture data over eastern CONUS generated through assimilating SMAP data into the Noah-MP land surface model

Sheng-Lun Tai, Zhao Yang, Brian Gaudet, Koichi Sakaguchi, Larry Berg, Colleen Kaul, Yun Qian, Ye Liu, and Jerome Fast

5 Pacific Northwest National Laboratory, Richland, 99352, USA

*Correspondence to*: Sheng-Lun Tai (sheng-lun.tai@pnnl.gov)

**Abstract.** An improved fine-scale soil moisture (SM) dataset at 1-km grid spacing, covering much of the eastern continental U.S., was generated by assimilating 9-km SMAP SM data into the v4.0.1 Noah-MP land surface model. With 12 ensemble members, the assimilation was carried out using the Ensemble Kalman Filter algorithm within NASA's Land Information System. The SM analysis for 2016 was fully validated against in-situ observations from four different networks and compared with four other existing datasets. Results indicate that this SM analysis surpasses other datasets in top-layer SM distribution, including a machine learning-based product, despite all SM estimates being less heterogeneous than observed. The analysis of anomalous errors suggests that large similarity in intrinsic errors is likely due to overlapping data sources among the selected SM datasets. More detailed evaluations were performed over two geographic areas. The observations collected by the Atmospheric Radiation Measurement facility in Oklahoma suggest that soil temperature and surface heat fluxes are concurrently simulated in good accuracy. Investigation into the 2016 southeastern U.S. drought response further indicates drier conditions and higher evapotranspiration estimates compared to GLEAMv4.1. Notably, large errors are associated with grids having clay soil textures, underscoring the need for refined model treatments for specific soil types to further improve SM estimates. The dataset is publicly available on Zenodo at https://doi.org/10.5281/zenodo.14370563.

## 1 Introduction

Soil moisture (SM) is a critical component in the complex interactions between the land surface and the atmosphere, influencing a range of processes that are vital for weather and climate dynamics. More specifically, it plays a significant role in regulating surface energy fluxes by controlling the partitioning of incoming solar radiation into sensible and latent heat 25 fluxes, thereby impacting atmospheric stability, boundary layer dynamics, and the initiation of convective systems (Dirmeyer et al. 2016; Ek and Holtslag 2004; Betts 2002; Taylor et al. 2011).

In addition to groundwater, precipitation falling onto ground surface contributes to the SM availability. Conversely, variations in SM heterogeneity can also influence the spatial and temporal distribution of precipitation through its effects on evapotranspiration rates and the atmospheric moisture and energy budgets (Katul et al. 2012; Hsu and Dirmeyer 2023). Hence,

the feedback loop between SM and precipitation is crucial for understanding and predicting regional hydrological cycles, droughts, and flood events (Koster et al. 2004; Dirmeyer et al. 2016). Furthermore, SM conditions can impact weather extremes such as heatwaves by modulating the surface energy balance and the efficiency of heat exchange between the land surface and the atmosphere (Seneviratne et al. 2010). These interactions occur across various spatial and temporal scales, underscoring the need for accurately capturing the spatial and temporal variabilities of SM distribution.

A variety of sensors such as Time Domain Reflectometry (TDR), capacitance probes, and neutron probes have been used in in-situ (ground-based) SM measurements. These measurements provide direct assessments of SM content at specific locations with high temporal resolution and accuracy in the soil column and are most useful for validating remote sensing data and calibrating hydrological models (Robock et al. 2000; Rasheed et al. 2022). However, their relatively sparse distribution hinders their applicability for characterizing realistic local-to-regional SM variability in broader regions despite efforts made
in expanding soil moisture observation networks (Diamond et al. 2013; Schaefer et al. 2007; Hawdon et al. 2014; Dorigo et al. 2021; McPherson et al. 2007; Wang et al. 2023). Conversely, remote sensing satellites such as the Soil Moisture Active Passive (SMAP) mission, Advanced Microwave Scanning Radiometer for the Earth Observing System (AMSR-E), Soil Moisture and Ocean Salinity (SMOS), and Sentinel-1 (Entekhabi et al. 2010; Njoku et al. 2003; Kerr et al. 2001; Torres et al. 2012), provide nearly global coverage of soil moisture estimates measured by passive and active microwave sensors. Passive
microwave sensors measure soil moisture based on microwave emissions from the Earth's surface, while active radar sensors use backscatter measurements to infer soil moisture levels (e.g., Kerr et al. 2001; Wagner et al. 2013). These satellite-based retrievals offer spatially extensive coverage and reasonable revisit times (1 − 3 days), contributing to large-scale hydrological and climate studies. Nevertheless, known uncertainties of satellite SM retrievals such as relatively coarse resolution [O (10 km)], limited accuracy (affected by vegetation, surface roughness, and temperature), shallow depth (only in depth of 0-5 cm
is measured), and environmental interference (rain, cloud, and snow cover) have posed challenges on their contributions to represent local-to-regional scale SM distribution (e.g., Colliander et al. 2017).

    Land Surface Models (LSMs) can simulate soil moisture conditions for any region by representing the interactions among the atmosphere, vegetation, and the ground (Niu et al. 2011; Lawrence et al. 2019; Liang et al. 1994). Key processes such as precipitation, infiltration, lateral flow, evaporation, plant transpiration, and groundwater table variations are parameterized in
LSMs. When precipitation occurs, water can infiltrate into the soil, accumulate, or run off, depending on soil characteristics and the rate of rainfall. Evaporation from the soil and transpiration from plants (collectively called evapotranspiration) reduce soil moisture, while infiltration and percolation move water downward through the soil profile. LSMs typically predict these processes to provide estimates of soil moisture at different depths over time. Various depths of soil layers can be configured to model the water movement between these layers in the soil column. A retrospective LSM simulation forced by observation-
constrained surface atmospheric conditions (rainfall, temperature, wind, humidity, and radiation, etc.), land and soil properties (leaf area index (LAI), albedo, land cover, soil texture, and permeability, etc.) is commonly used to reproduce the soil

conditions. Despite the advantages, state-of-the-art LSMs still contain uncertain, incomplete, and/or unresolved physical processes that may introduce biases into the simulated land surface properties.

As a way to mitigate such modelling issues, data assimilation (DA) techniques such as ensemble Kalman filter (EnKF), variational methods (e.g., 3DVar and 4DVar), and Bayesian approaches have been used to merge multiple sources of observational data (in-situ measurements and satellite retrievals) with LSM simulations to optimize soil moisture simulations through improving initial conditions and parameter estimates, enhancing the accuracy of soil moisture predictions and hydrological forecasts (e.g., Reichle et al. 2002; Crow and Wood 2003; Kumar et al. 2008; Chao et al. 2022; Martens et al. 2017). In any DA approach, the assimilation scheme must be coupled with an LSM. As such, the generated analysis consists of model states which are always physically balanced and can be directly used as the initial conditions of LSM. Some additional advantages of utilizing DA techniques in generating high-resolution SM data include their flexibility in data resolution (output frequency, horizontal grid spacing, and vertical layers) and domain coverage, the possibility to incorporate any improvements in the coupled models and/or new observables, and the availability of the full suite of land surface properties relevant for studies of atmospheric boundary layer and hydraulic processes.

A variety of satellite soil moisture retrievals have been assimilated into different LSMs. For example, Draper et al. (2012) assimilates data measured by both active microwave advanced scatterometer (ASCAT) and passive AMSR-E into the Catchment model. Liu et al. (2010) also assimilates ASCAT and AMSR-E, but the Noah LSM was chosen as the core model. Seo et al. (2021) conducted experiments over CONUS, which assimilates SMAP and ASCAT data into the Joint UK Land Environment Simulator (JULES) using the local ensemble transform Kalman filter (LETKF). They found SMAP data are more beneficial than ASCAT in terms of improvement in soil moisture estimate. Mousa and Shu (2019) assessed the potential impacts of assimilation of SMAP, SMOS, and ASCAT on spatial representation in soil moisture over Africa and reported that SMAP has overall superior performance compared to SMOS and ASCAT. Earlier studies have specifically explored the impact of SMAP soil moisture data assimilation soil moisture estimates, hydrological modelling, and drought monitoring across different regions of the globe. For example, studies have shown promising results by assimilating SMAP soil moisture data into the Noah-MP land surface model (e.g., Rouf et al. 2021; Ahmad et al. 2022). The Noah-MP LSM is widely used in both research and operational systems (e.g., Ma et al. 2017; He et al. 2023; Johnson et al. 2023). Compared to its predecessor (Noah LSM), Noah-MP introduces substantial improvements that enhance realism, flexibility, and process representation. Those updates include dynamic vegetation models, multi-layer snow and soil physics, stomatal resistance schemes, and canopy-interception processes. Given the compatibility with coupled models, Noah-MP is integrated into state-of-the-art atmospheric models including WRF, making it an excellent choice of LSM to study land-atmosphere coupling processes. Research by Rouf et al. (2021) discussed how the spatial resolution of SMAP SM data (36-km versus 9-km) and the grid spacing of analysis (12.5 and 0.5 km) would impact SM estimation over Oklahoma using the framework of NASA's Land Information System (LIS). They showed the accuracy in SM analysis is enhanced when assimilating the 9-km SMAP data with 0.5 km LSM grid spacing. Likewise, Yin and Zhan (2020) showed a positive influence of soil moisture data assimilation coupled with Noah-

95 MP simulations in the continental U.S. (CONUS) and underscores the need for fine-scale soil moisture data to achieve an optimal result. Ahmad et al. (2022) further demonstrated the positive impact of SMAP DA on soil moisture estimate in South Asia along with sensitivities to SMAP data bias correction settings. In Chakraborty et al. (2024), an improved soil moisture distribution over India was obtained by incorporating SMAP soil moisture into the Indian Land Data Assimilation System (ILDAS).

Emerging higher-resolution (i.e., 1 km) soil moisture datasets such as SMAP-derived 1-km downscaled surface soil moisture data (Fang et al. 2022) and Sentinel-1 surface soil moisture data (Fan et al. 2025), could potentially provide finer-scale soil moisture information and may be incorporated into data assimilation processes. However, as described in Fang et al. (2022), the spatial coverage and availability of the downscaled SMAP dataset is notably reduced compared to the 9-km dataset. On the other hand, although there are multiple studies demonstrating the impacts of assimilation of Sentinel-1 data (e.g., Brocca

et al. (2024), Filippucci et al. (2022), Foucras et al. (2020), Gao et al. (2017), and Meyer et al. (2022)), the experiments were all performed over smaller and more localized areas as opposed to the more extensive domains used in SMAP-based studies. This is primarily due to the contrasts in sensor characteristics between these two satellites. While Sentinel-1 measures at a higher spatial resolution (~ 1 km) than SMAP, it has relatively lower radiometry sensitivity, much longer revisit time (3 to 4 times), and requires more complex preprocessing. This means Sentinel-1 may be less sensitive to subtle differences in soil

moisture content than SMAP and would be unlikely to capture day-to-day variability. Hence, in general, SMAP data is more suitable for regional/global scale applications than Sentinel-1.

 Building upon these studies, we aim to improve local-to-regional soil moisture distributions over much of the east CONUS region by assimilating the SMAP Level 3 (L3) 9-km soil moisture product into the 1-km grid spacing Noah-MP LSM. While earlier studies (e.g., Rouf et al. (2021)) choose not to use higher resolution precipitation forcing data, we use the 4-km NCEP

Stage IV Quantitative Precipitation Estimate (QPE) data (Lin and Mitchell (2005)) as the LSM's precipitation forcing. The Stage IV product has been a unique precipitation dataset since it takes advantage of both the weather radar and rain gauge observation networks over the CONUS to robustly reconstruct precipitation distribution. Moreover, instead of only focusing on the importance of SM DA for improving soil moisture estimates as other studies did, we also explore how SM DA may influence other simulated land surface properties on both seasonal and regional bases. We assess the performance of our dataset

over the full study domain but also explore key regions in additional detail. Specifically, we evaluate the performance of our dataset in a known "hotspot" of land-atmosphere coupling using the dense in-situ observations collected by the Oklahoma Mesonet and DOE's Atmospheric Radiation Measurement (ARM) facility in Southern Great Plains (SGP), which were also used in the study of Rouf et al. (2021). In addition, we examine our dataset's characterization of the extreme drought conditions affecting the southeast U.S. over the fall and winter of 2016. Comparisons are made to alternative SM data sets, including

datasets generated through machine learning approaches, to better understand the value of DA incorporated with an LSM. The primary goal of this study is to demonstrate the development of a year-long soil moisture dataset for the eastern U.S., which

can also be readily used in land–atmosphere coupled simulations by providing the essential boundary conditions needed for model initialization.

The remaining parts of this manuscript are organized as follows: the analysis domain and period are described in Section 2. The methodologies and datasets employed in this study are detailed in Section 3. The results of the impact of SM data assimilation and the evaluations of the generated SM estimate along with the other existing SM datasets are discussed in Section 4. Lastly, the summary and discussion are provided in Section 5.

## 2 Analysis domain and period

Our study domain encompasses a wide swath of the central and eastern CONUS (Figure 1). The time period for the analysis covers the entire year of 2016 from January 1 through December 31, 2016. This analysis period was selected in order to complement land-atmosphere coupled simulations associated with the 2016 Holistic Interactions of Shallow Clouds, Aerosols, and Land-Ecosystems (HI-SCALE) field campaign (Fast et al. 2018). The locations of in-situ measurements from the United Sates Climate Reference Network (USCRN), Soil Climate Analysis Network (SCAN), Oklahoma Mesonet (OKMet), and ARM SGP are overlaid on the map in Figure 1a. The soil texture and land cover maps are given in Figure 1b and 1c, respectively. Table 1 summarizes the grid numbers and their percentages over the study domain, for each classification of soil texture and land cover. The top three soil types (besides water) are silt loam (24.02%), loam (18.88%), and sandy loam (15.7%), whereas grassland, cropland, and cropland/natural vegetation mosaics are top three land cover types accounting for 22.2, 19.64, and 10.2%, respectively, of points in the domain.

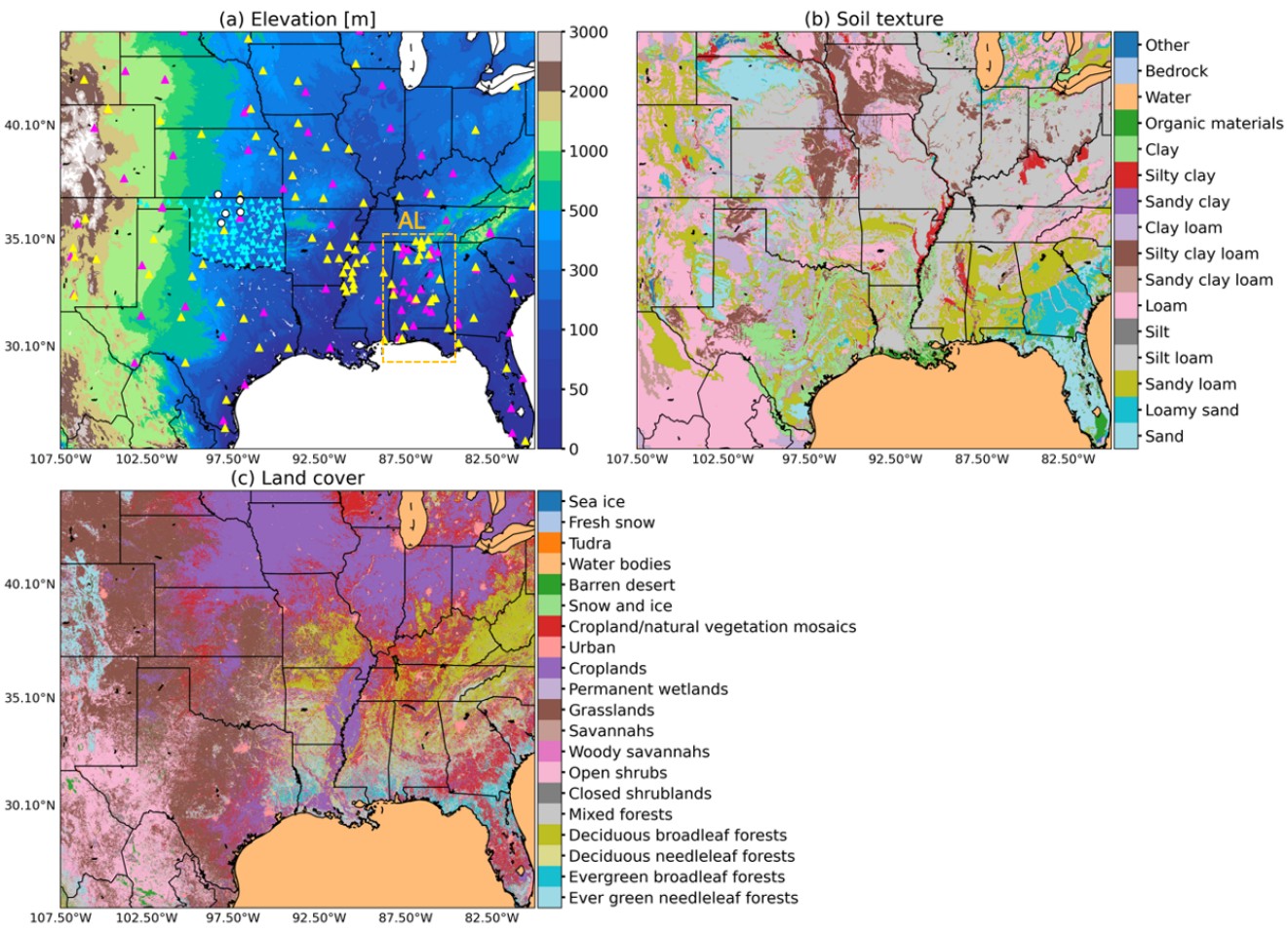

**Figure 1** Maps illustrating the study domain over eastern CONUS. The yellow, magenta, and cyan triangles denote the stations of SCAN, USCRN, and OKMet observational networks, respectively. The white circles mark the locations of selected ARM SGP sites. The domain soil texture was categorized into 14 soil types (**c**) according to the NCEP/STATSGO+FAO classification. The domain land cover comprised 18 main types based on the MODIS-derived IGBP classification. The subdomain AL is denoted by the orange box (dashed line) in (a).

**Table 1** Summary of grid number and percentage of total grids for the soil texture/land cover types.

| Soil texture | | | Land cover | | |
|---|---|---|---|---|---|
| Class | # of grids | Percentage of total grids [%] | Class | # of grids | Percentage of total grids [%] |
| Sand | 370,227 | 7.12 | Evergreen needleleaf forests | 120,816 | 2.32 |

| | | | | | |
|---|---|---|---|---|---|
| Loamy sand | 140,072 | 2.69 | Evergreen broadleaf forests | 66,193 | 1.27 |
| Sandy loam | 816,849 | 15.70 | Deciduous needleleaf forests | 344 | 0.007 |
| Silt loam | 1249,730 | 24.02 | Deciduous broadleaf forests | 373,716 | 7.18 |
| Loam | 982,206 | 18.88 | Mixed forests | 332,114 | 6.38 |
| Sandy clay loam | 41,522 | 0.80 | Closed shrublands | 23,319 | 0.45 |
| Silty clay loam | 240,916 | 4.63 | Open shrubs | 514,667 | 9.89 |
| Clay loam | 187,450 | 3.60 | Woody savannahs | 100,390 | 1.93 |
| Silty clay | 63,818 | 1.23 | Savannahs | 27,746 | 0.53 |
| Clay | 205,676 | 3.95 | Grasslands | 1154,805 | 22.20 |
| Organic materials | 39,598 | 0.76 | Permanent wetlands | 9,591 | 0.18 |
| Water | 842,008 | 16.19 | Cropland | 1021,681 | 19.64 |
| Other | 22,069 | 0.42 | Urban | 64,997 | 1.25 |
| | | | Cropland/Natural vegetation mosaics | 521,029 | 10.02 |
| | | | Snow and ice | 33 | 0.0006 |
| | | | Barren desert | 28692 | 0.55 |
| | | | Water bodies | 842,008 | 16.19 |

# 3 Methodology and datasets

## 3.1 NASA Land Information System and Noah-MP land surface model

The NASA Land Information System (LIS) is an advanced modelling and data assimilation framework designed to better simulate land surface processes and improve our understanding of terrestrial hydrology, biogeochemistry, and climate interactions (Kumar et al. 2006; Peters-Lidard et al. 2007). LIS incorporates multiple hydrological and LSMs and data assimilation techniques to optimize the representation of land surface processes. This model-observation integration enhances the accuracy and reliability of simulations by leveraging the strengths of different models and observational datasets. It is functionable in assimilating satellite-derived observations of soil moisture, vegetation dynamics, and other land surface variables to improve the initialization and calibration of model simulations. Its versatility and scalability make it suitable for

both research and operational uses. Given the above, LIS is primarily used in this study to generate realistic representation in soil states through assimilation of SMAP soil moisture retrievals into Noah-MP land surface model.

The version 4.0.1 Noah-MP LSM (Ek et al. 2003; Niu et al. 2011; Yang et al. 2011) was run within LIS to simulate the relevant land surface processes across the study domain. The Noah-MP model was run with a 0.01° by 0.01° horizontal grid
spacing and using a 15-min time step. The specific model configurations utilized are detailed in Table 2. Each soil column within the study region is represented by four layers with depths of 10, 30, 60, and 100 cm below the ground surface. The surface soil moisture updates are transmitted to deeper layers according to model formulations in water diffusivity and hydraulic conductivity. More specifically, while moisture fluxes between successive layers controls how water moves within each soil column, excess water above saturation in any layer will be transferred to the next unsaturated layer downward. The
Noah-MP LSM can be driven by many sources of meteorological forcing data as desired. Note that external irrigation and groundwater extraction were not explicitly simulated in Noah-MP and these processes might be important for certain locations (Yang et al. 2020, 2021).

**Table 2** Selected parameters, parameterizations, and forcing data used in the configured Noah-MP LSM.

| LSM parameter/parameterization/forcing data | |
|---|---|
| Land cover | MODIS (IGBP-NCEP) (Friedl et al. 2002) |
| Elevation, slope, and aspect | SRTM30-v2.0 (Farr et al. 2007) |
| Greenness | National Center for Environmental Prediction (Gutman and Ignatov 1998) |
| Vegetation | Dynamic vegetation option |
| Maximum albedo | National Center for Environmental Prediction (Robinson and Kukla 1985) |
| Canopy stomatal resistance | Ball-Berry method (Ball et al. 1987) |
| Snow surface albedo | Canadian land surface scheme (Verseghy 1991) |
| Runoff and groundwater | Simple groundwater model, SIMGM (Niu et al. 2007) |
| Surface-layer drag coefficient | General Monin-Obukov similarity theory (Brutsaert 1982) |
| Snow and soil temperature | Semi-implicit option |
| Partitioning of rain and snowfall | Jordan91(Jordan 1991) |
| Lower boundary of soil temperature | Noah native option |
| Supercooled liquid water and frozen soil permeability | NY06 (Niu et al. 2007) |
| Surface meteorological forcing | NLDAS-2 and Stage IV QPE (precipitation) |

 ## 3.2 Datasets

The datasets employed in this study include the forcing data that drive the Noah-MP LSM (section 3.2.1 – 3.2.3), multiple in-situ observations (section 3.2.4) used as the benchmarks for intercomparison among our SM estimate and the other existing SM datasets (section 3.2.5 – 3.2.7).

### 3.2.1 Enhanced SMAP Level 3 soil moisture data

The Soil Moisture Active-Passive (SMAP) uses passive (radiometer) L-band microwave remote sensing to estimate land surface soil moisture and freeze/thaw state (O'Neill et al., 2014). The L-band radiometry offers all-weather, diurnal sensing of the surface dielectric properties which are a function of the near-surface soil moisture. The SMAP has a 2- to 3- day revisit frequency and two overpasses (morning and afternoon) at local time 6 a.m. and 6 p.m., respectively. One of the SMAP products, the enhanced SMAP Level 3 soil moisture product (SPL3SMP_E; O'Neill et al., 2020), is primarily used for assimilation in this study. It consists of daily estimates of global soil moisture within the top soil layer ($\sim 5$ cm depth) on a cylindrical 9-km Equal-Area Scalable Earth Grid (https://nsidc.org/data/spl3smp_e/versions/6), spanning from 31 March 2015 to present.

### 3.2.2 North America Land Data Assimilation System Phase 2 (NLDAS-2)

The NLDAS-2 (Xia et al. 2012) aims to provide high-resolution, near-real-time and retrospective datasets that integrate land surface model outputs with observations to monitor and simulate land surface conditions across North America. It is available at hourly intervals and on a 12.5-km spatial grid from January 1979 to present. A wide range of land surface variables such as soil moisture, soil temperature, snow cover, evapotranspiration, and runoff are provided. Meteorological forcing variables such as precipitation, temperature, wind speed, and solar radiation are also included. The NLDAS-2 is used in this study as the meteorological forcing data to drive the Noah-MP LSM.

### 3.2.3 NCEP Stage IV Quantitative Precipitation Estimate

The NCEP Stage IV Quantitative Precipitation Estimate (QPE) (Lin and Mitchell (2005)) is a high-resolution, quality-controlled dataset produced by the National Centers for Environmental Prediction (NCEP). It integrates precipitation data from multiple sources, including NEXRAD radar, rain gauges, and satellite observations, to provide accurate and detailed precipitation estimates across the contiguous United States. With a grid spacing of 4 km at hourly intervals, Stage IV QPE is widely used in meteorology, hydrology, and climate research for tasks such as weather forecasting, flood modelling, and studying precipitation trends. We replace the precipitation data in the NLDAS-2 by the Stage IV QPE data as it not only provides a higher-resolution and more realistic precipitation forcing over the CONUS region but also improved SM estimates

in our test simulations. As an example, the comparison of instantaneous rain rate obtained from NLDAS-2 and Stage IV precipitation at 00 UTC of August 30, 2016 (Figure S1) demonstrates that the Stage IV data provides more heterogeneous precipitation distribution than NLDAS-2 over the study domain.

### 3.2.4 In-situ measurements

In-situ soil moisture observations used in this study were obtained from the 1) U.S. Climate Reference Network (USCRN); 2) Soil Climate Analysis Network (SCAN); 3) Oklahoma Mesonet (OKMet, McPherson et al. 2007); 4) ARM SGP (Sisterson et al. 2016). The USCRN and SCAN data are acquired from the International Soil Moisture Network (Dorigo et al. 2021). The four networks are selected as the benchmarks of our SM analysis due to either their relatively wide spatial coverages or preferred site locations. Besides atmospheric and environmental parameters such as air temperature, humidity, and wind conditions, both SCAN and USCRN stations are equipped with sensors that measure critical soil parameters, including soil moisture and temperature at the depths of 5, 10, 20, 50, and 100 cm. The USCRN and SCAN are superior among available soil moisture networks as many of their stations (112 and 91 sites from USCRN and SCAN, respectively) are uniformly distributed over the study domain (Figure 1). They are used to evaluate our SM analysis along with other existing SM datasets (Table 3). The OKMet (120 sites) and ARM SGP (6 sites) observations are adopted as their site locations are densely distributed (average distance between any two stations is shorter than 30 km) over a portion of the Southern Great Plains (SGP) region which is one of the hotspots with strong land-atmosphere coupling (e.g., Fast et al. 2018; Sakaguchi et al. 2022). In addition to SM, the soil temperature observations and the latent and sensible heat fluxes measured by the Soil Temperature and Moisture Profiles (STAMP) and Eddy Correlation Flux Measurement System (ECOR) deployed by the ARM SGP facility, are also used to concurrently assess the simulated soil properties and surface heat fluxes. Note soil moisture (temperature) measured at a depth of 5 cm below ground surface was primarily used to compare with the model-estimated surface soil moisture (soil layer depth = 0 to 10 cm).

### 3.2.5 ERA5-Land reanalysis

The ERA5-Land (Muñoz-Sabater et al. 2021) is a global reanalysis dataset that provides essential land variables with a grid spacing of 0.1 degree and is valid at hourly frequency, spanning from January 1950 to present. It is continuously produced by rerunning the land component (Tiled ECMWF Scheme for Surface Exchanges over Land incorporating land surface hydrology (H-TESSEL)) of the ECMWF ERA5 climate reanalysis that sequentially assimilates available meteorological observations (Hersbach et al. 2020). Despite model uncertainties due in part to imperfect atmospheric forcing, unresolved physical processes, and lack of observational constraint, the spatiotemporal coverages of ERA5-Land dataset have been advantageous in many land surface applications including flood or drought monitoring and forecasting. It is thus employed in this study as one of the SM reference data, providing more insights though the comparison.

### 3.2.6 Global Land Surface Satellite soil moisture (GLASS SM)

The global, daily 1-km GLASS soil moisture product (GLASS SM; Zhang et al. 2023) was derived using an ensemble learning model (eXtreme Gradient Boosting – XGBoost) that integrates multiple datasets as the machine learning (ML) model's inputs, including the remotely sensed Global Land Surface Satellite (GLASS) products (Liang et al. 2021), ERA5-Land reanalysis products (Muñoz-Sabater et al. 2021), and static auxiliary datasets (e.g., Multi-Error-Removed Improved-Terrain (MERIT) and Global gridded soil information (SoilGrids; Poggio et al. 2021). The ground-based soil moisture archived by the International Soil Moisture Network (ISMN) and the 0.25° grid spacing combined soil moisture data of European Space Agency's Climate Change Initiative (ESA CCI; Dorigo et al. (2017)) are collectively used as the target data of training in ML. The validations carried out for the GLASS SM product in Zhang et al. (2023) demonstrated its capability in capturing temporal dynamics of measured soil moisture. Hence, given its novelty in the methodology and high spatial resolution (1km), the GLASS SM data is used as one of the benchmarks in this study.

### 3.2.7 Global Land Evaporation Amsterdam Model (GLEAM)

The GLEAM (Global Land Evaporation Amsterdam Model; Miralles et al., 2011) is a state-of-the-art dataset that provides global estimates of soil moisture, terrestrial evaporation (or evapotranspiration), and related hydrological components. GLEAM soil moisture data is derived from satellite observations and model simulations. It integrates a variety of satellite observations and meteorological data, such as soil moisture from microwave remote sensing, vegetation indices, and meteorological data of precipitation, air temperature, and radiation. The version 4.1 of GLEAM (Miralles et al. 2025) is used in our analysis, which is available at 0.1-degree resolution between the period of 1980 to 2023.

**Table 3** Soil moisture estimates analyzed in this study.

| Soil moisture product | Grid spacing | Spatial coverage | Temporal resolution | Temporal coverage | References |
|---|---|---|---|---|---|
| SPL3SMP_E | 9 km | Global | Daily | 31 March 2015 - present | O'Neill et al. (2020) |
| ERA5-Land | 0.1° | Global | Hourly | 1950 - present | Muñoz-Sabater et al. (2021) |
| GLASS SM | 1 km | Global | Daily | 2000 - 2020 | Zhang et al. (2023) |
| GLEAM v4.1 | 0.1° | Global | Daily | 1980 - 2023 | Miralles et al. (2025) |
| SMAPDA | 1 km | East CONUS | 6 hourly | 2016 | - |

### 3.3 Open loop and data Assimilation simulations

The open loop simulation (named "OL" hereafter) employs configuration noted in Table 2 and spins up between January 1, 2011 and March 31 of 2015. It refers to the integration of Noah-MP land surface model without any assimilation of external observations. The long spin-up period (greater than 4 years) ensures that soil states reach the equilibrium state before conducting data assimilation (Cosgrove et al. 2003; Rodell et al. 2005). Since the SMAP SM data is only available from March 31, 2015 onwards, the DA simulation started on 00 UTC of April 1, 2015 and ended on 00 UTC of January 1, 2017. The ensemble Kalman filter (EnKF) assimilation algorithm implemented in the LIS is utilized to assimilate the SMAP SM retrievals into the Noah-MP-modelled estimates. The EnKF's sequential assimilation algorithms including two main steps (model propagation and data assimilation update) are coupled with model integration and executed recursively. Here, the Noah-MP is the nonlinear forward model to advance the propagation step and generate the prognostic state vector forward in time. The update step occurs whenever any observations are valid, and the update of prognostic state variable can be described by the equation below:

$$\hat{x}_{k+1}^a = \hat{x}_{k+1}^b + K\left(y_{k+1} - H_{k+1}\left(\hat{x}_{k+1}^b\right)\right). \tag{1}$$

Where $\hat{x}_{k+1}^a$ stands for the analyzed (updated) state of variable $x$ at time step $k + 1$. $\hat{x}_{k+1}^b$ represents the background state of variable $x$ integrated from time step k. The Kalman gain matrix $K$ and the innovation vector $\left(y_{k+1} - H_{k+1}\left(\hat{x}_{k+1}^b\right)\right)$ are required when updating the background state. Here, $y_{k+1}$ denotes the observations valid at time step $k + 1$ and $H_{k+1}$ is the observation operator that applies conversion and interpolation in time and space to the model state variable in order to conform with the observable.

The ensemble simulations are required at each propagation step to provide an estimate on the model spread (uncertainty). Here, the NASA Land Data Toolkit (LDT; Arsenault et al. (2018)) is used to initialize the ensemble simulations based on the OL simulation restart output file at 2345 UTC on March 31, 2015. The initial conditions of those ensemble members are obtained by perturbing atmospheric forcing variables as listed in Table 4. Perturbation type is grouped as either multiplicative (M), sampled from a log-normal distribution or additive (A) which is sampled from a normal distribution.

**Table 4** Description of parameters used in meteorological forcing perturbations for the ensemble simulations

| Perturbed meteorological forcing | Perturbation type | Standard deviation | Cross-correlations with perturbations | | | |
|---|---|---|---|---|---|---|
| | | | SW | LW | P | $T_{air}$ |
| Shortwave radiation (SW) | M | 0.2 W m$^{-2}$ | 1.0 | -0.3 | -0.5 | 0.3 |
| Longwave radiation (LW) | A | 30 W m$^{-2}$ | -0.3 | 1.0 | 0.5 | 0.6 |

| Precipitation (P) | M | 0.5 mm | -0.5 | 0.5 | 1.0 | -0.1 |
| Near-surface air temperature ($T_{air}$) | A | 0.5 K | 0.3 | -0.6 | -0.1 | 1.0 |

According to the sensitivity study regarding the impact of ensemble size in Ahmad et al. (2022), the ensemble spread (measured by standard deviation across all members) may be flattened when the number of replicates increases beyond 15. We experimented with 12 and 24 ensemble members, and the result suggested minor difference is demonstrated in terms of soil moisture representation. Hence, the DA experiment we show here has an ensemble size of 12. The model and SMAP soil moisture retrieval error standard deviations are set as $0.04\,\mathrm{m^3\,m^{-3}}$. Due to the existence of relative systematic difference between SMAP and modelled SM, the cumulative distribution function (CDF) matching technique (Reichle and Koster 2004) is used for bias correction of the SMAP soil moisture retrievals using Noah-MP model data as the reference. Monthly CDFs of the SMAP soil moisture retrievals and the Noah-MP-simulated soil moisture were both generated using the NASA LDT and used to map the SMAP SM retrievals into the Noah-MP-modelled soil moisture space prior to assimilation. The reference period for the monthly matching is two years in total, ranging from Jan. 1, 2015, to Dec. 31, 2016. Since the SMAP SM data is representative of the top soil layer ($\sim$5 cm deep from surface), the topmost soil layer soil moisture is employed as the model state variable during assimilation. The DA simulation as well as its SM data are abbreviated as "SMAPDA" hereafter. More detailed discussion regarding its performance in estimated SM is covered in the Section 4.

### 3.4 Metrics for DA impact measuring and evaluation

### 3.4.1. Soil moisture analysis increment

To assess the impact of the SMAP SM data assimilation on the soil moisture estimates, we analyze the soil moisture analysis increments generated from the DA experiment (SMAPDA). The analysis increment refers to the difference between the analysis (optimized estimate of the state after DA) and the background forecast (model state before DA). It is a measure of how much the model state has been corrected (updated) by incorporating new observations, which is not only related to the deviation from model background to observation but also modulated by observation and model errors. In the EnKF approach, the model error varies in time and space and is estimated using the ensemble spread (standard deviation of ensemble simulations). We use the cumulative number and temporal mean of soil moisture analysis increments to indicate the spatial distribution of observational constraint by the SMAP_L3_E data and highlight the areas that experience an overall wetting or drying due to the cycling of assimilation. Note the SMAP_L3_E data was subset to hourly data and assimilated when it matches the model time step.

### 3.4.2. Evaluation against in-situ measurements

The soil moisture estimates generated through different approaches are evaluated against in-situ measurements using the metrics of anomaly correlation coefficient (ACC), root-mean-square error (RMSE), and Bias defined as follows:

$$ACC = \frac{\sum_{i=1}^{n}(P'_i - \bar{P'})(M'_i - \bar{M'})}{\sqrt{\sum_{i=1}^{n}(P'_i - \bar{P'})^2 \sum_{i=1}^{n}(M'_i - \bar{M'})^2}} \tag{2}$$

$$RMSE = \sqrt{\frac{1}{N}\sum_{i=1}^{n}(P_i - M_i)^2} \tag{3}$$

$$Bias = \frac{1}{N}\sum_{i=1}^{n}(P_i - M_i) \tag{4}$$

Where $P$ represents the estimated top layer soil moisture, and $M$ stands for corresponding in-situ soil moisture measurement. $N$ is the total number of selected samples. The $ACC$, ranging from -1 to 1, measures how well the temporal anomalies (departures from the monthly mean) of two time series (model estimates $P'$ and observation $M'$) match each other. Here in Eq. (2), it is essentially computed as the Pearson correlation coefficient using the estimated and observed soil moisture anomaly time series in each location. Since soil moisture timeseries has strong seasonal cycles, the removal of seasonal signal when computing ACC helps quantify the skill in capturing soil moisture temporal variations across all time scales. The ACC is commonly used to verify the impact of soil moisture data assimilation due to the necessity in isolation of seasonal cycle which is highly consistent between open loop and assimilation experiments (e.g., Kumar et al. 2009). While RMSE [Eq. (3)] is used to measure the mean difference between the modeled and in-situ SM, Bias [Eq. (4)] is computed as the overall deviation (including the signs) of the modeled SM from in-situ SM observations. In addition, the standard deviation (STD) is also calculated for each SM dataset to quantify the spatial heterogeneity in SM across the given sites at different locations:

$$STD = \sqrt{\frac{1}{N}\sum_{i=1}^{n}(S_i - \bar{S})} \tag{5}$$

Here, $S_i$ refers to individual SM data points and $\bar{S}$ stands for mean over the entire dataset.

## 4 Results

### 4.1 Increments from SMAP soil moisture data assimilation

To gauge how much observational information was effectively assimilated into the model, we examined the outputs of SM analysis increments at the top layer (5-cm depth). Figure 2 illustrates the maps of cumulative number (hours) of SM analysis increment over each of the three-month long periods. Overall, the SMAP SM data assimilation is more effective in spring, summer, and fall (Figures 2b, 2c, and 2d) as opposed to winter (Jan-Feb-Mar; Figures 2a). The relatively small number

of analysis increment shown in the Jan-Feb-Mar period (Figure 2a) is likely due to the increased uncertainty in L-band microwave radiometer SM retrieval as a result of snow cover and frozen ground in the cold season (e.g., Liu et al. 2021). While analysis increments are distributed over the majority of domain, there are grids that received zero update, especially in the eastern part of the domain. In the default setting, the LIS would only assimilate observations where the SMAP data's retrieval quality is flagged as successful. Those zero-update pixels are most likely covered by dense vegetation. As such, the sensitivity of surface SM is usually distinctly reduced and thus flagged as unsuccessful retrievals. Nevertheless, despite generally less effective assimilation over this region, a few spots in Florida and partially Georgia and South Carolina show most frequent updates from DA across the entire domain.

Figure 3 demonstrates the spatial distribution of mean SM analysis increments over the four seasons. The calculation of mean increment only includes samples with non-zero increments. While consistently positive increments are shown in Texas and northern Mexico throughout the year, seasonal variations are evident in portions of the Great Plains. For instance, in Kansas, more negative (positive) increments are seen for Jan-Feb-Mar/Apr-May-Jun (Figures 3a and 3b) and Jul-Aug-Sep/Oct-Nov-Dec (Figures 3c and 3d), respectively. This suggests that compared to SMAP observations, the model most likely has a consistent dry bias over part of Texas and the adjacent Mexican territory, and the biases are more variable temporally in other parts of the domain including the northern SGP. One possible cause of those DA increments is due to the model deficiencies and missing physical processes. For instance, SMAP detects increased soil moisture which may be partially due to irrigation, whereas the Noah-MP LSM used in this study does not explicitly simulate irrigation (e.g., Felfelani et al. (2018) and Lawston et al. (2017)). In addition, practices like tilling or cover cropping affect surface moisture and are likely not captured by the model physics. Other missing/unrealistic model treatments in runoff schemes, dynamic groundwater level, seasonal varying vegetation and root systems may also modulate the increment patterns. Moreover, biases in meteorological forcing (e.g., radiation, temperature, and winds) may also affect how evapotranspiration is estimated and thus the soil moisture. SMAP data assimilation often compensates for these errors, especially after dry spells or in transition seasons (spring and fall).

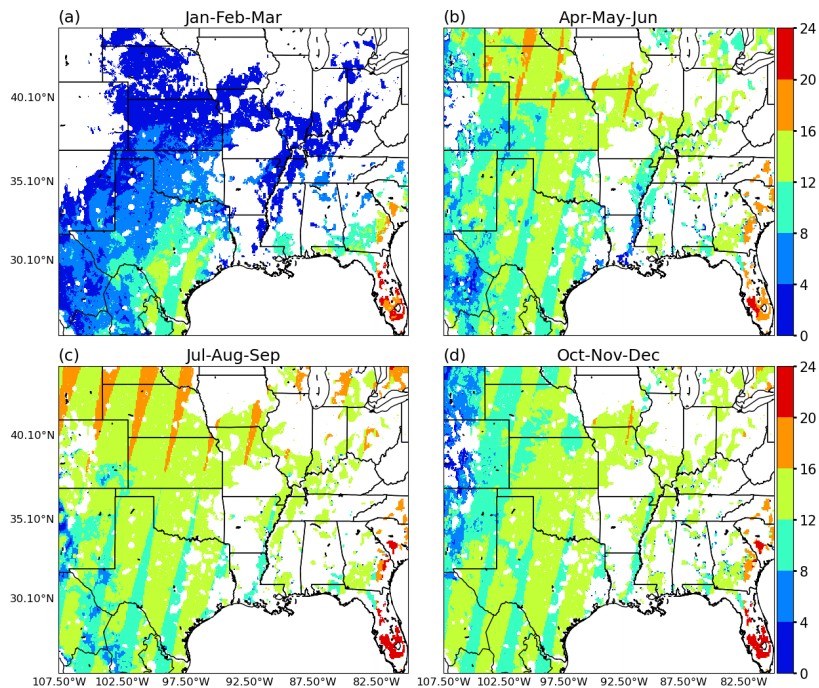

**Figure 2** Maps of cumulative number of DA SM increments computed for the periods of (a) Jan-Feb-Mar, (b) Apr-May-Jun, (c) Jul-Aug-Sep, and (d) Oct-Nov-Dec in 2016.

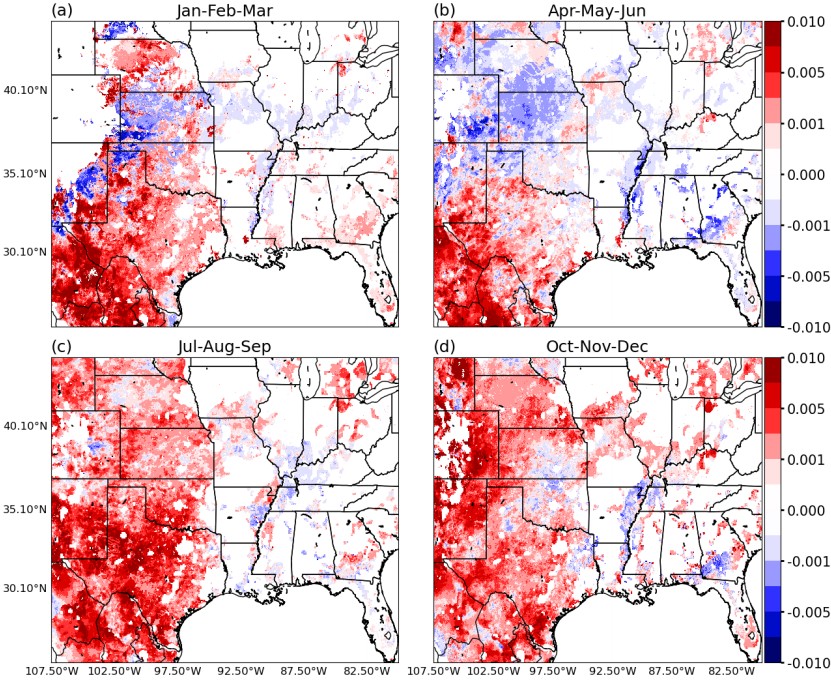

**Figure 3** Similar to Figure 2, but for mean SM increments.

## 4.2 Comparison with existing surface SM datasets

To assess the performance of our SM analyses (SMAPDA and OL) along with other existing SM products, we conduct a comprehensive intercomparison among all derived datasets (Table 3) against a collection of in-situ measurements from four observational networks (USCRN, SCAN, OKMet, and ARM SGP). Note the assessments were conducted separately against the USCRN and the SCAN datasets despite both have well-distributed site locations over the study domain. This was carried out purposely in order to verify whether any inconsistency between their instruments and/or measurements may alternatively bias the validation results. The following sub-sections discuss the evaluation results referenced by using the observations from each network.

### 4.2.1 Evaluation using USCRN soil moisture observations

SM estimates from SMAPDA, OL, GLASS SM, ERA5-Land, GLEAMv4.1, and SMAP AM (the morning overpass of SMAP_L3_E) are first evaluated using the in-situ observations from the USCRN (Figure 1). The metrics described in Section 3.4.2 are computed accordingly. Since only SMAPDA and ERA5-Land consist of SM representations through the entire soil column, surface (top-layer) SM representations are primarily assessed here. To perform one-to-one comparisons with in-situ data, for each SM product, the daily SM timeseries data at the grid cells closest to the observational site locations are extracted. The 2-D histograms as given in Figure 4 are illustrated to visualize the differences between the observations and the estimates and depict the contrasts among the datasets. All scatter points are grouped by 50 bins (2-D pixels) and the contours are smoothed using the Gaussian filter for an improved visualization. The more samples concentrated along the diagonal line, the better estimate it would be considered. (placeholder for rootzone SM evaluation)

The results indicate SMAPDA has the highest ACC (~ 0.8) among all SM estimates. While it's slightly higher than OL in general, Figure 5b shows SMAPDA improves over OL at most of the sites, indicating SM DA does optimize the SM dynamics despite overall minor differences in RMSE and Bias. It also shows that SMAPDA's RMSE and Bias (0.085 and 0.005 $m^3$ $m^{-3}$) are slightly larger than what GLASS SM has (0.083 and -0.004 $m^3$ $m^{-3}$). Since the GLASS SM uses in-situ observations including those from USCRN as the target when training the ML model (i.e., not independent), it is not surprising the GLASS SM magnitudes better align with the USCRN data in general. However, the GLASS SM has the smallest ACC (0.574) compared to all other estimates. This implies the ML algorithm may not capture the temporal evolution of SM but only the instantaneous SM values due to the selected variables (i.e., absolute SM) in the cost function. At any rate, these two 1-km grid spacing products significantly outperform others. Constructed in 0.1-degree grid spacing, both ERA5-Land and GLEAMv4.1 (Figures 4c and 4f) may partially suffer from a relative coarser resolution in addition to differences in treatments in physical processes. As a result, their RMSEs are all greater than 0.1 $m^3$ $m^{-3}$. Meanwhile, relatively larger biases are also computed (0.017 and -0.006 $m^3$ $m^{-3}$). The SMAP AM also has poor skill in SM estimation given its highly scattered samples in the 2-D

histogram despite relatively low bias. Although it has the second highest ACC among all (0.625), it still under-represents temporal variability at site-level when comparing with SMAPDA (Figure 5e).

We also note the cut-in (smallest) values of surface SM vary notably across the SM products. For example, the GLEAMv4.1 and SMAPDA have relative larger cut-in values of ~0.05 to 0.06 $m^3m^{-3}$. Whereas the ERA5-Land and SMAP AM are valid above approximately 0.02 $m^3 m^{-3}$. The GLASS SM has negligible limit on the smallest SM value. The differences in these cut-in SM values may be associated with either the formulations of land surface models or the observational sensitivities and could at least partially affect how good each estimate agrees with the observation

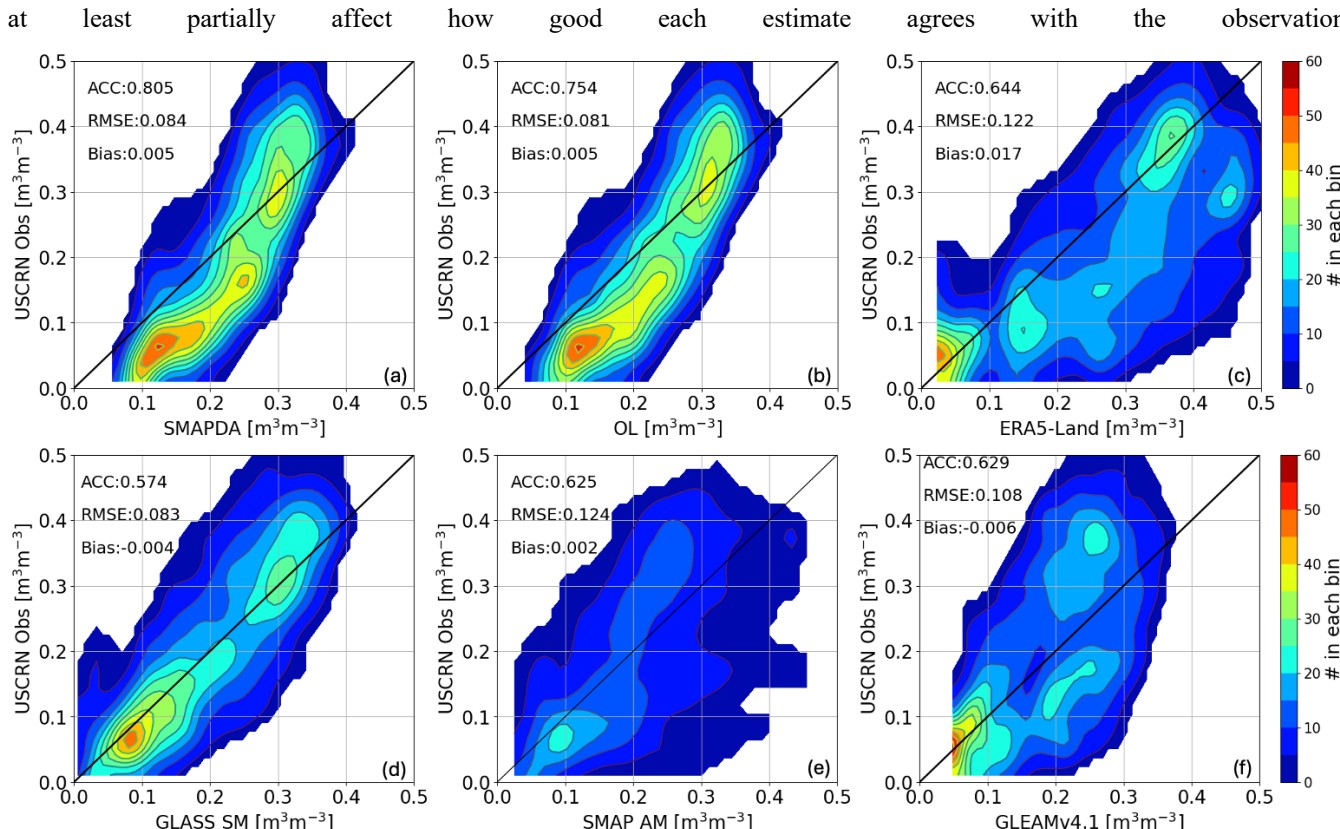

**Figure 4** 2-D histograms summarizing the evaluation results using the observational data measured by the USCRN network. Panels (a) to (f) represent results of SMAPDA, OL, ERA5-Land, GLASS SM, SMAP AM, and GLEAMv4.1. 50 bins are used to generate the 2D histograms. Anomaly correlation coefficient (CC), RMSE, and Bias are given in the upper left corner of each panel.

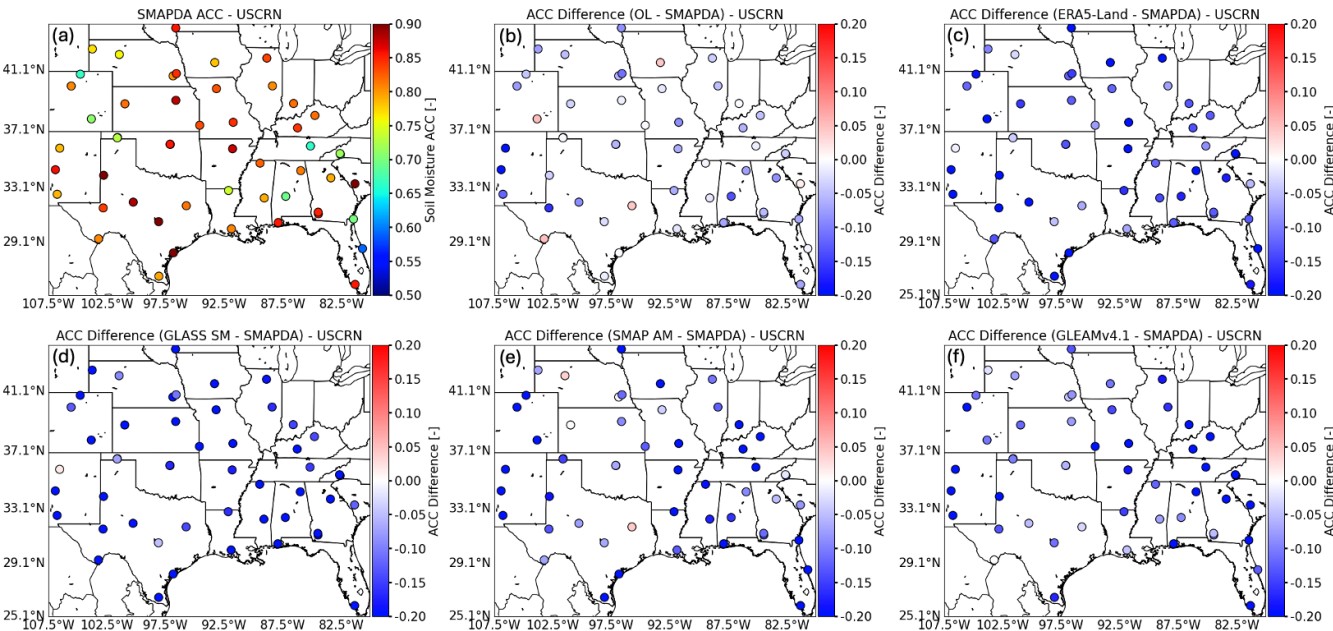

Figure 5 (a) Site-wise SM ACC computed for SMAPDA using the USCRN observations. The ACC differences subtracting SMAPDA results from (b) OL, (c) ERA5-Land, (d) GLASS SM, (e) SMAP AM, and (f) GLEAMv4.1 are also illustrated.

Figure 6a illustrates the disaggregated RMSEs for SMAPDA at each USCRN site. The RMSE differences between SMAPDA and other estimates are given in Figures 6b – 6f to better visualize relative performance. Not surprisingly, relatively minor differences between OL and SMAPDA are analyzed (Figure 6b). While the RMSE differences are rather mixed between SMAPDA and GLASS SM (Figure 6d), distinct and extensive increases in RMSEs are observed in the cases of ERA5-Land, SMAP AM, and GLEAMv4.1 (Figures 6c, 6e, and 6f), especially for those sites in the southeast U.S., and coastal sites in Florida and Texas.

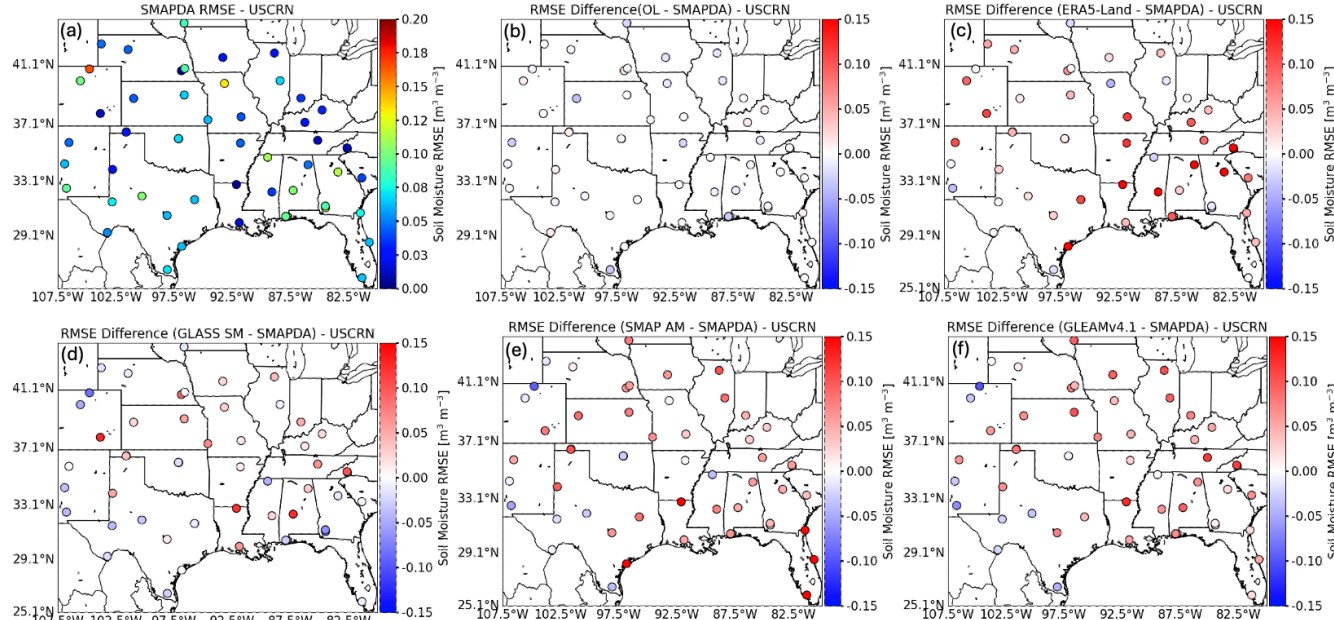

**Figure 6** Similar to Figure 5, but for SM RMSE.

Likewise, the biases are displayed in Figure 7. In all SM datasets, wet bias is more evident in the southeastern U.S. sites than others, whereas dry bias is distinct across many sites in the northern and eastern Great Plains despite variability in their magnitudes. This consistent bias pattern implies that these SM estimates may share common sources of uncertainties in

observational data and/or treatments in the models. Further model improvements may be carried out to focus on the correction of this common issue.

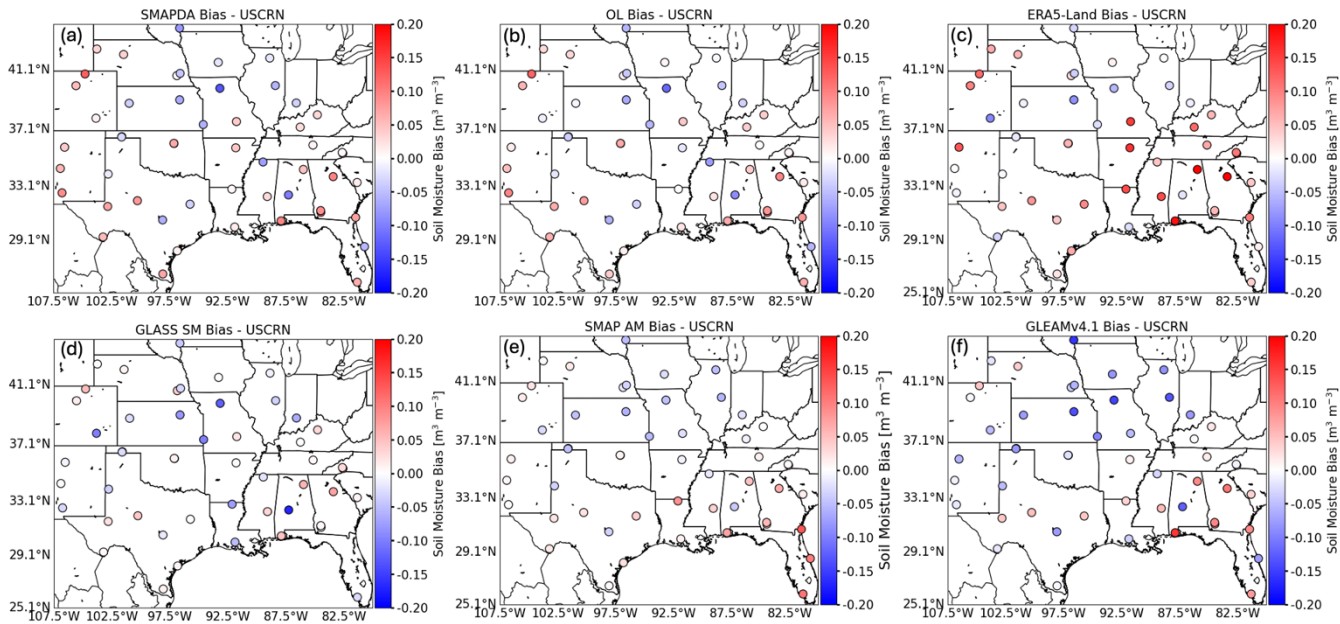

**Figure 7** Site-wise Bias computed using the USCRN observations. Results for (a) SMAPDA, (b) OL, (c) ERA5-Land, (d) GLASS SM, (e) SMAP AM, and (f) GLEAMv4.1 are illustrated.

To further examine the potential errors in common among the six SM estimates, we calculated the RMSE anomaly for each dataset. The RMSE anomaly is obtained by subtracting annual mean RMSE from the daily timeseries of each estimate. It extracts intrinsic variation in SM errors from the original SM timeseries and thus facilitate bias-free intercomparison. A diverse variation among the datasets is shown in Figure 8. Despite relatively large day-to-day variability in the SMAP AM timeseries than other datasets, the multiday variability in SMAP AM is similar to GLEAMv4.1. For example, both of them

show much larger SM errors from January to April and relatively smaller errors present in late spring and summer. The errors climb when it transitions into late fall and early winter. There is also much similarity between the ERA5-Land and GLASS SM timeseries. Despite minor discrepancies, compared to other datasets, they both show relatively smaller variation over the one-year period with slightly larger errors in April, June, and July. These results are not surprising as SMAP data is one of the ingredients of GLEAMv4.1 (Miralles et al. 2025), whereas GLASS SM adopts ERA5-Land soil moisture as the SM input data

in their ML model (Zhang et al. 2023). Figure 8 also indicates that SMAPDA demonstrates a unique trend with the smallest errors before June and peak errors occur in early July and late November.

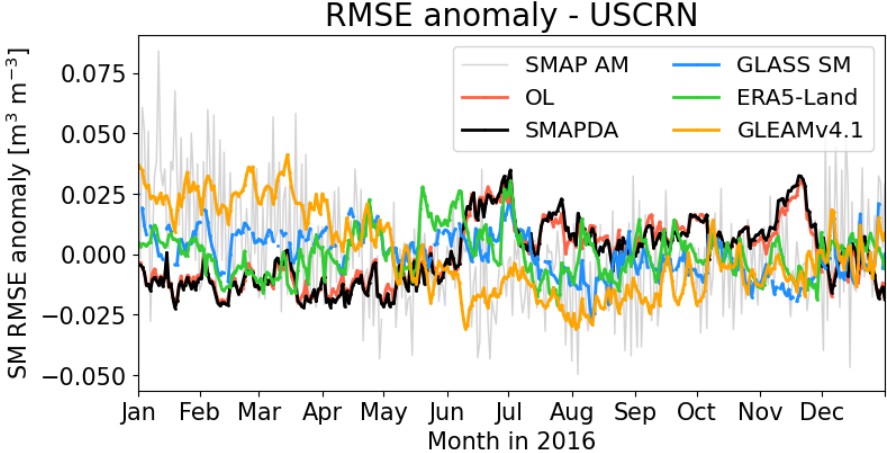

**Figure 8** The SM RMSE anomaly timeseries computed against USCRN observations during 2016. Results of SMAP AM, OL, SMAPDA, ERA5-Land, GLEAMv4.1, and GLASS SM are denoted by colored lines as indicated in the legend.

### 4.2.2 Evaluation using SCAN soil moisture observations

Along the same line as discussed in Section 4.2.1, we examined the SM 2-D histograms as referenced by the SCAN observations (Figure S2). Overall, similar conclusions can be drawn from the comparisons, implying that the evaluation is robust with very little dependence on selected reference SM observations. For example, SMAPDA further improves ACC on top of OL (Figures S1a and S1b), showcasing the positive impact on capturing SM variability in time. Moreover, SMAPDA and GLASS SM remain the top two performers among the SM products in terms of low RMSE (0.089 and 0.095 $m^3 m^{-3}$) and Bias (0.001 and -0.006 $m^3 m^{-3}$). But again, despite better alignment with the diagonal in general, GLASS SM has much smaller ACC than SMAPDA. It also suggests GLASS SM has more off-diagonal samples than the SMAPDA, likely due in part to the used of ERA5-Land as the initial SM guess for GLASS SM. Since ERA5-Land has relatively scattered samples in the 2-D histogram (Figure S2c) and ML algorithm does not overfit by design (Zhang et al. 2023), some pixels may receive less correction than others. The estimate from GLEAMv4.1 (Figure S2f) suffers from generally smaller SM estimates (capped around ~ 0.38 $m^3 m^{-3}$), which potentially causes severe underestimation. SMAP AM has the least bias among all estimates (Figure S2e). However, it also owns many samples far off the diagonal, which lower the overall skill scores. As a result, their RMSEs are all greater than 0.11 $m^3 m^{-3}$, which is about 20% more than what is computed for SMAPDA.

The site-wise ACC and RMSEs given in Figures S3 and S4 confirm that SMAPDA is the top performer among the SM estimates in general as it shows consistency in producing more realistic temporal evolution and relatively small error across the SCAN sites. Excessive SM errors (positive differences) are found at several sites in the southeastern U.S. when evaluating the ERA5-Land and SMAP AM. Despite small contrast in RMSE differences across the SCAN sites, the estimates from GLEAMv4.1 show extensively larger errors at more locations, leading to non-trivial mean RMSE of 0.112 $m^3 m^{-3}$. Figure S5

shows that, regardless of the in-situ observations, the bias pattern of each SM estimate resembles those given in Figure 7, suggesting that the analyzed biases should be rather robust and representative.

Figure S6 shows that the SM RMSE anomaly of SMAPDA is very similar to those of GLASS SM (light blue) and even SMAP AM (light gray) when assessed using the SCAN data. The ERA5-Land and GLEAMv4.1 exhibit trends partially
different from the other three estimates. Specifically, ERA5-Land (GLEAMv4.1) has relatively smaller (larger) errors than the other three estimates from January to April and tends to produce rather larger (smaller) errors from October to December. Despite the differences in these results and those from the comparison against USCRN (Figure 8), similarities in RMSE anomalies among the analyses remain clear. This is most likely due to the various SM estimates using duplicate sources of SM data, even though different methods are employed to arrive at the final estimates.

**4.2.3 Regional assessment over the Southern Great Plains**

The Southern Great Plains (SGP), including Oklahoma, has been recognized as one of the hotspots for strong land-atmosphere coupling (LAC; Santanello et al. (2009); Tao et al. (2019)). Earlier studies revealed the key physical processes that modulate the strength of LACand how LAC influences the lifecycle of convective clouds using model and observational datasets generated for this region. For instance, Fast et al. (2018) investigated the impact of SM spatial heterogeneity on
simulated convective clouds near the ARM SGP site using a large-eddy simulation model for a selected event during the 2016 HI-SCALE field campaign. They found that the scales of SM gradient in the model can significantly affect the presence of simulated cloud populations even with identical atmospheric conditions. Sakaguchi et al. (2022) further analyzed the LES model data produced by Fast et al. (2019) using the spectral analysis and demonstrated the SM spatial heterogeneity may strengthen secondary circulations and extend their spatial scales. Both studies concluded that a more realistic and high-
resolution representation of SM is desired to better understand LAC at local-to-regional scales (~1 km and greater). This motivates us to examine how SMAPDA SM estimate perform in this region in comparison to other datasets and the evaluations are carried out by leveraging highly concentrated observations measured by the OKMet (Figure 1).

As shown in Figure 9, the performance of GLASS SM degrades when evaluated against the OKMet data. The 2D histogram shows most samples occur in the bins above the diagonal, meaning that GLASS SM (Figure 9d) generally
underestimates SM (mean Bias: -0.038 $m^3$ $m^{-3}$). Whereas when comparing with data from USCRN and SCAN (Figures 4d and S1d), GLASS SM has much better agreement with the observations. This is most likely due to the exclusion of OKMet data in their ML training process. Since ML is purely data-driven, the skill of ML-based SM estimate highly depends on the availability of in-situ observations. Conversely, SMAPDA exhibits the lowest annual mean RMSE (0.087 $m^3$ $m^{-3}$) in general in comparison to the other four datasets (0.106, 0.107, 0.104, and 0.112 $m^3$ $m^{-3}$ for ERA5-Land, GLASS SM, SMAP_AM, and
GLEAMv4.1, respectively). The annual mean RMSE for SMAPDA stays close to the RMSEs obtained when comparing against the USCRN and SCAN observations (0.084 and 0.0894 $m^3$ $m^{-3}$, respectively). This demonstrates that a physically

constrained model tend to perform more consistently and mitigate soil moisture biases despite uncertain and neglected/unresolved physical processes in the model.

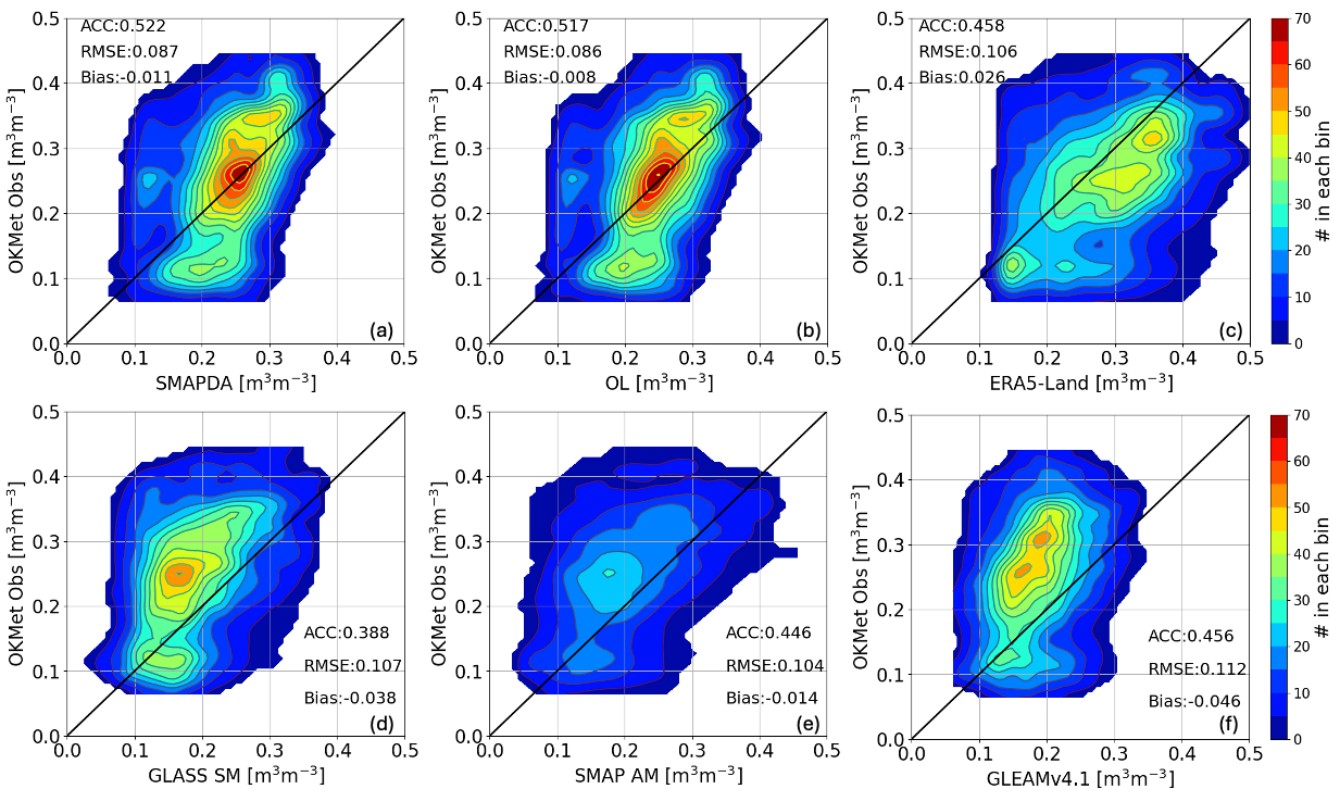

**Figure 9** Similar to Figures 4 and S1, but for results computed against the OKMet observations.

The relatively dense spatial distribution of the OKMet sites enables further investigation into the realism of estimated SM spatial heterogeneity. We computed daily standard deviation (STD) across all OKMet sites for each SM estimate as a way to quantify the spatial SM heterogeneity (meaning how spread the SM values are in space). Figure 10 shows that observed STD (magenta) is mostly larger than what is estimated by any of the derived SM approaches over the year despite notable day-to-day variations. Even though SMAPDA and GLASS SM top the others in SM estimates based on the evaluations shown earlier, they both underestimate the SM spatial heterogeneity with an averaged STD ~ 0.6 $m^3$ $m^{-3}$, which is about 25% less than observed. GLEAMv4.1 and SMAP AM have even smaller STDs over the period. While ERA5-Land tends to have larger and more comparable variances as observed, it does not accurately distribute those SM values in space (Figure S7).

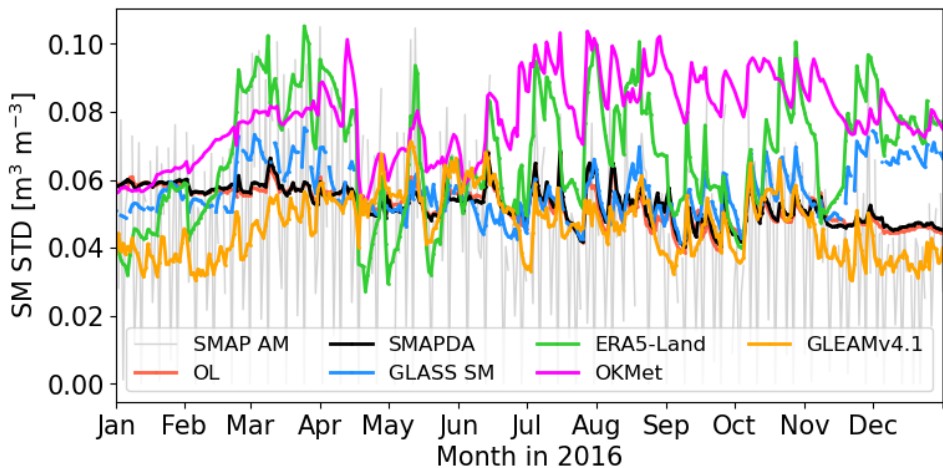

**Figure 10** Daily timeseries of SM standard deviation across OKMet sites computed for each SM estimate.

In addition to soil moisture, the ARM SGP facility (through instruments of STAMP and ECOR) has collected soil temperature and surface heat flux (latent and sensible) measurements that are critical for land-atmosphere coupling research as these quantities directly modulate the strength of turbulent mixing in the atmospheric boundary layer. Here we primarily assess how SMAPDA represents SM, soil temperature (ST), latent heat flux (LHF), and sensible heat flux (SHF) using the concurrent measurements collected across six ARM SGP sites (E31, E33, E37, E38, E39, and E41) as denoted in Figure1a. Note SM and ST data from the STAMP are not valid in January.

Results show that SMAPDA reproduces the observed monthly trends in SM and slightly overestimates SM (annual mean model-observation difference of + 0.04 $m^3$ $m^{-3}$) with larger positive biases in winter months (Figure 11a). The annual mean ST is warmer in SMAPDA than observed (difference: + 0.84 K) which can be attributed to relatively distinct warm bias in summer months (June – September) (Figure 11b). While LHF has an annual mean difference of -12.14 W $m^{-2}$ when compared to the observations (Figure 11c), it is considered minor as annual LSM error can be approximately - 20 W $m^{-2}$ based on earlier studies (ARM 2014). Whereas in the case of SHF, SMAPDA tends to underestimate for most of the months (Figure 11d). This is likely due to consistent wet bias in SM throughout the year (Figure 11a), leading to increased energy partition in latent heat and corresponding reduction in sensible heat. Distinct positive biases even appear over summer months (June and July) despite higher simulated ST (Figure 11b).

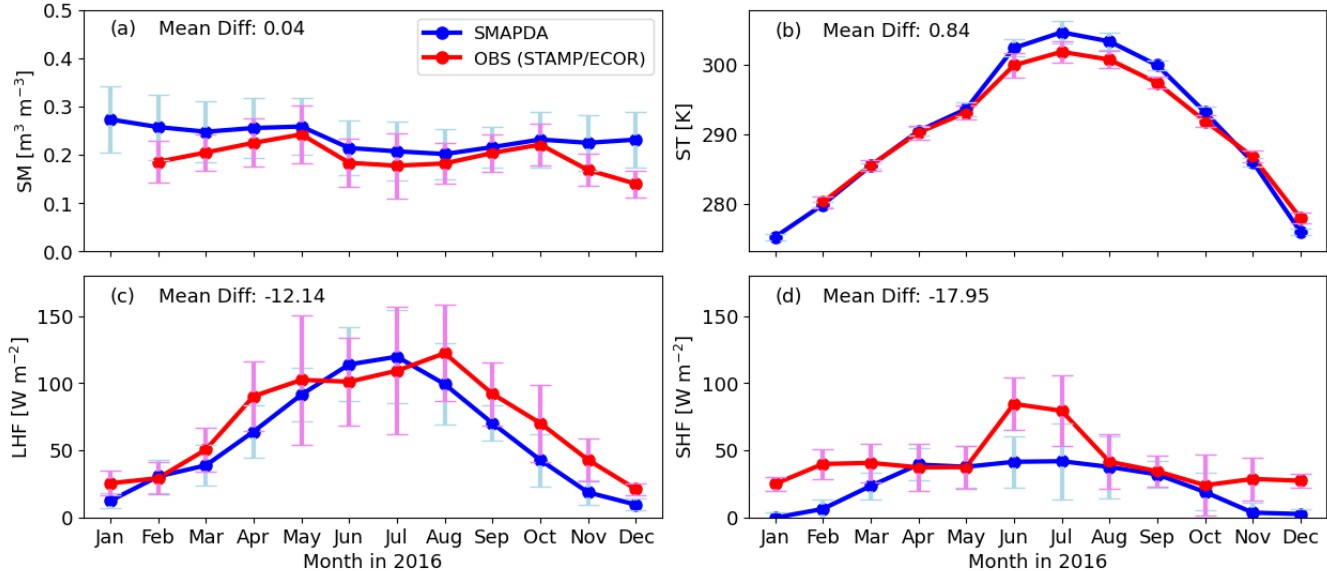

**Figure 11** Timeseries of monthly mean of (a) SM, (b) ST, (c) LHF, and (d) SHF. Blue (red) line with circles represents results obtained from SMAPDA simulations (ARM SGP observations [STAMP/ECOR]) as computed across six ARM SGP sites (Figure 1a)

### 4.2.4 Regional analysis associated with 2016 drought in the Southeastern U.S.

The southeastern U.S. experienced one of the most significant drought events in the region during the fall of 2016 (peaked in October and November) based on historical record (Park Williams et al. 2017). It primarily affected parts of Georgia, Alabama, Tennessee, and the Carolinas. The drought reached extreme and exceptional levels, especially in northern Georgia and Alabama, where some areas experienced their driest conditions in history. A combination of factors, including below-average rainfall during the spring and summer months and unusually high temperatures led to the increased evaporation and reduced soil moisture and thereby the drought conditions in fall. The drought severely impacted agriculture, leading to reduced crop yields, and contributed to widespread wildfires in the Appalachian region. The strained water resources also posed a great challenge on water availability for communities and industries. Hence, we aim to explore the representativeness of the SMAPDA SM estimate under the extreme drought conditions in the southeastern U.S.

A subdomain "AL" which covers Alabama (as denoted in Figure 1a) was chosen for conducting the following analyses since the in-situ measurements from USCRN and SCAN are relatively denser in Alabama than in other areas in the southeastern U.S. (Figure 1a). In addition to the spatial variability of soil types (Figure 12a), the relatively large fraction of forests cover (Figure 12b) can further complicate how soil moisture is distributed through hydraulic processes such as evapotranspiration (ET), interception, infiltration, runoff, groundwater recharge, and hydraulic redistribution due to the presence of root systems

and tree canopies. Here, we selectively examine the relationship between SM and ET under the drought conditions by comparing the SMAPDA output with the GLEAMv4.1 data. The GLEAMv4.1 was chosen for this specific comparison as it is the only product among all benchmark datasets that concurrently provides SM and observation-constrained ET representations.

To assess whether the two datasets (SMAPDA and GLEAMv4.1) represent the drought event, we first use the 5-year (2012 - 2016) soil moisture data from OL simulation to infer monthly climatological mean and standard deviation at each pixel over the area. Based on the climatological baseline, the standardized soil moisture anomaly (SMA) can then be computed to better quantify the severity of drought conditions. According to Ontel et al. (2021), Jiménez-Donaire et al. (2020), and Tian et al. (2022), when SMA is between 0 and -1, the drought is considered mild drought. Moderate drought occurs if SMA lies between -1 and -2. Lastly, when SMA falls below -2, severe drought condition is defined. It shows both SMAPDA and GLEAMv4.1 suggest moderate to severe drought conditions occurred in this region during the months of September, October, November (Figure 13). Due to the contrast in sample size, SMAPDA demonstrates more spatial heterogeneity in SMA compared to GLEAMv4.1. Otherwise, the result reconfirms the drought period as defined in other relevant studies.

The monthly statistics from both SMAPDA and GLEAMv4.1 are given in Figure 14. Both estimates exhibit a decreasing trend in SM over the summer months (JJA) as well as a steeper decline in fall (SON) over the AL subdomain (Figure 14a). Except in winter months (DJF) where the SM estimate is slightly larger in SMAPDA than in GLEAMv4.1, soil conditions produced by SMAPDA are consistently drier than GLEAMv4.1. Both datasets suggest increases in ET before June with similar magnitudes (Figure 14c), mostly due to the seasonal increase in the solar insolation as well as the leaf area. However, in summer (JJA), SMAPDA produces much larger ET than GLEAMv4.1 does, which leads to much drier soil conditions concurrently. This then facilitates the intensification of drought conditions in the fall, leading to further reduction in water availability through the soil columns which significantly limits the amount of ET as opposed to GLEAMv4.1.

Although in-situ ET observations are not available through the USCRN and SCAN measurements, the SM observations (Figure 14b) suggest while SMAPDA overall captures how SM evolves over time, GLEAMv4.1 gives much weaker responses to the SM drying process than SMAPDA. This ultimately produces overall much larger wet bias in SM for GLEAMv4.1 than for SMAPDA in the fall. Whether data from all grid cells in the AL subdomain (Figures 14a and 14c) or only the 17 grid cells nearest to the in-situ measurements (Figures 14b and 14d) are used, the trends in both SM and ET are very similar. This suggests that the evaluations illustrated in Figures 14b and 14d are representative for the subdomain. As we investigate in more detail through comparison at each individual site (Figure 15), we find most of the large errors in SMAPDA's SM estimate can be attributed to sites' soil properties (Figure 15), specifically where the clay soil types are present (site #3 and 5: Clay; site# 7: Silty clay). At those sites, the enhanced temporal variability in SM is distinct but underestimated by both SMAPDA and GLEAMv4.1 estimates. This suggests both approaches are unable to capture the drastic changes in SM, likely due in part to the nature of clay soil texture. The conclusion regarding soil texture-dependent errors seems to hold even when we extend the analysis to all sites (USCRN and SCAN) located in the study domain (Figure S8). The relatively weaker dependency on

landcover types illustrated in Figure S9 confirms that the errors are much more sensitive to soil textures than landcover types. Overall, it shows although the sample sizes vary among soil types, the soil moisture error remains relatively higher at sites with clay soil than other soil types. This echoes what was reported in Colliander et al. (2022) stressing further model refinements may be needed to improve treatments in resolving hydraulic processes for the variants of clay soil.

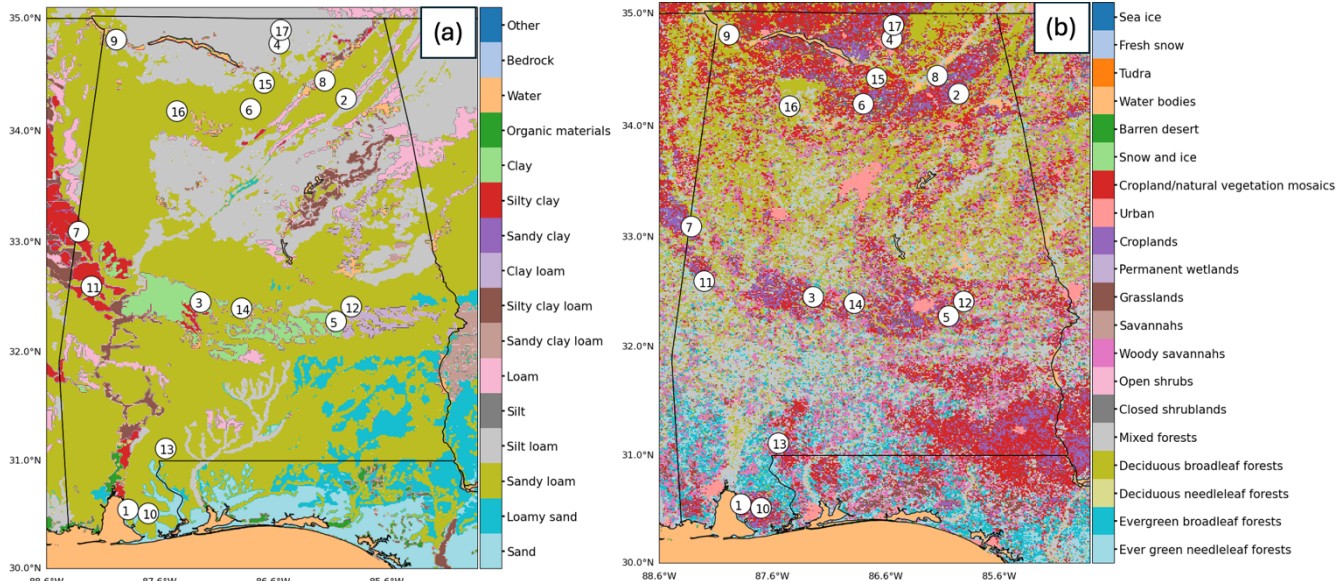

**Figure 12** The zoomed-in maps of (a) soil and (b) land cover types over the AL subdomain as marked in Figure 1a. The locations of 17 valid observational sites from USCRN and SCAN are denoted by the white circles with numbers overlaid in correspondence of panels in Figure 13.

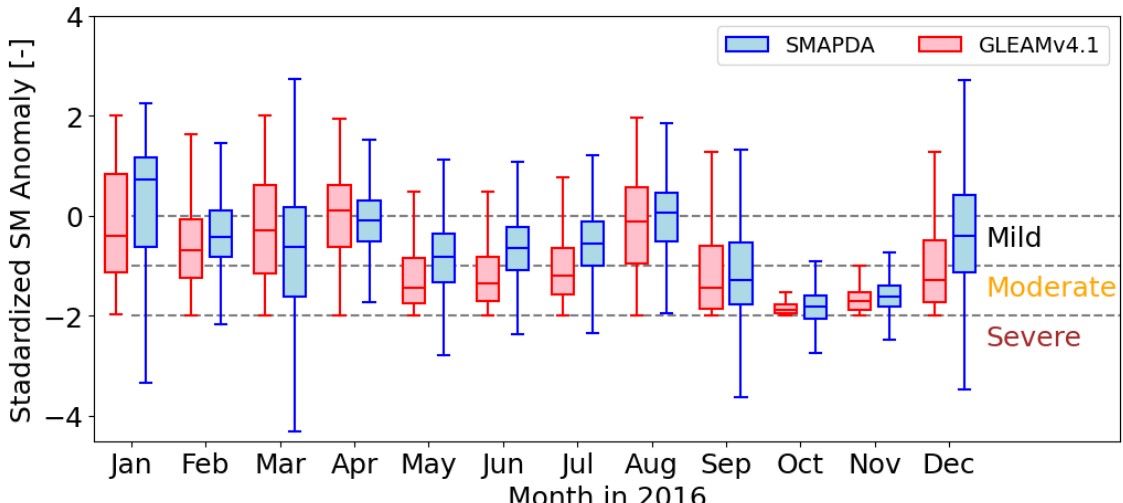

**Figure 13** Box and whisker plot for monthly standardized SM anomaly (SSA) computed for SMAPDA and GLEAMv4.1 data in 2016 referenced by their own climatology (2012 - 2016). Dashed lines denote the thresholds for the defined drought conditions.

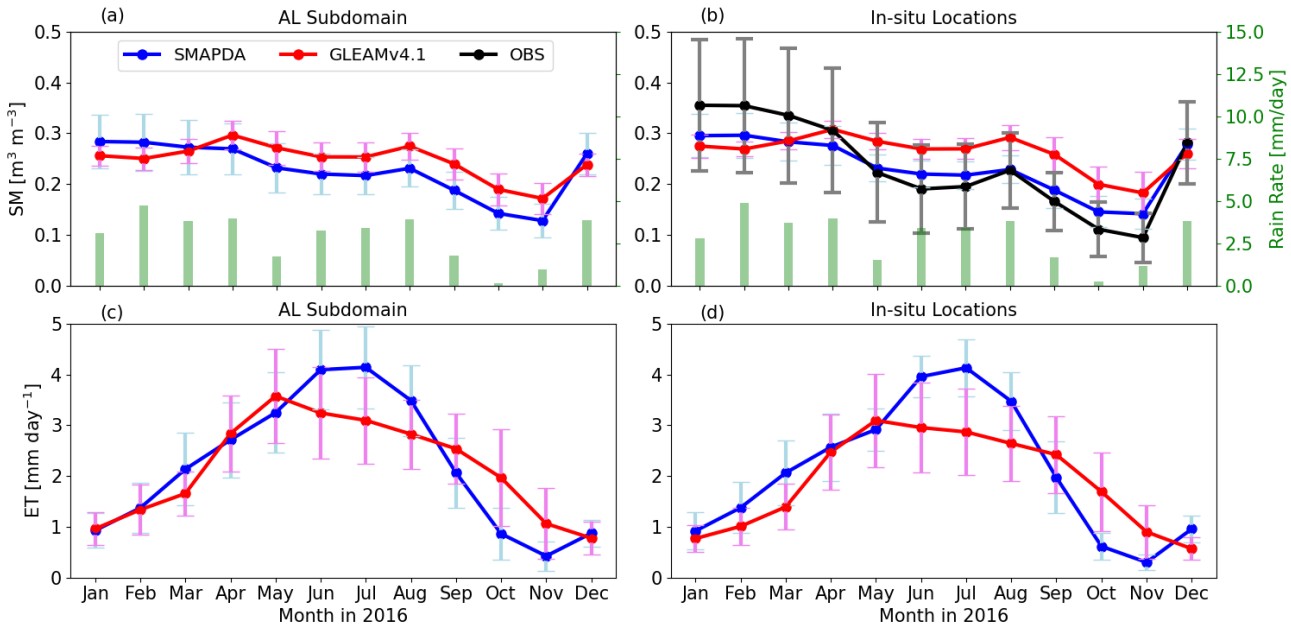

**Figure 14** Monthly mean and standard deviation (denoted by error bars) of SM (a, b) and ET (c, d) over the AL subdomain and among 17 in-situ observation locations as denoted in Figure 12. The mean rain rate represented by green bars in (a) and (b) are computed from SMAPDA correspondingly.

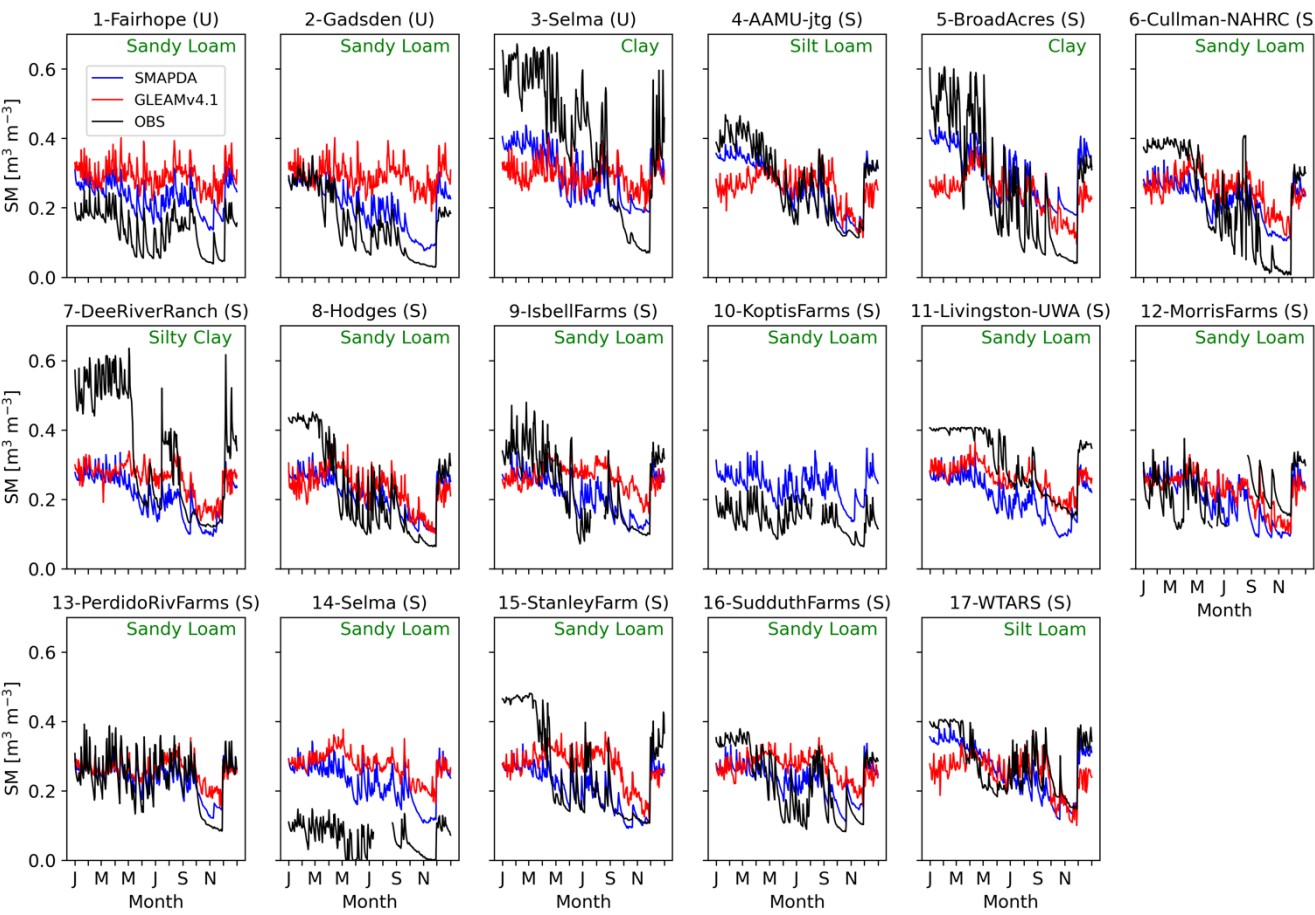

**Figure 15** SM daily timeseries comparison at each in-situ observation location. The numbers given in the title above each panel correspond to the locations as marked by the numbered white circles in Figure 12. The site names and the corresponding observational networks (as indicated by either U (USCRN) or S (SCAN) in the parenthesis) are readable from the titles. Soil texture type is indicated by green texts in the top right corner of each panel.

## 5. Summary and discussion

To facilitate an improved representation of local-to-regional scale SM distribution, we generated a high-resolution SM dataset at a 1-km grid spacing by assimilating the 9-km SMAP SM data into the Noah-MP land surface model. The dataset has a spatial coverage over the east CONUS and has frequency of 6 hours for the entire 2016. The SMAP SM data assimilation is accomplished under the framework of NASA's Land Information System using the EnKF algorithm. In the DA simulation, 12 ensemble members were initialized by perturbing the selected variables in meteorological forcing data (NLDAS-2 and Stage IV). The subset of daily SMAP SM overpasses is assimilated hourly when applicable. The generated SM estimate is comprehensively assessed by using the in-situ SM observations collected in the networks of USCRN, SCAN, OKMet, and

ARM SGP and compared with the performance of other existing SM datasets such as the morning overpass of SMAP (SPL3SMP_E) data, ERA5-Land, GLASS SM, and GLEAMv4.1.

Overall, the evaluation result suggests the resulting soil moisture estimate through DA, which we refer to as SMAPDA, exhibits the top performance among the examined datasets. It improves SM temporal variability in most of the evaluated sites when comparing with the estimate from the OL simulation (experiment without DA). While the SMAPDA and GLASS SM are considered the top two SM estimates based on the skill metrics computed against USCRN and SCAN observations (e.g., CCs are ~0.8 and ~0.7 and RMSEs are ~0.08 and ~0.09 $m^3\ m^{-3}$, respectively), SMAPDA surpasses GLASS SM when validated

against OKMet data (independent observations for both SMAPDA and GLASS SM). Being a fully data-driven ML product, the GLASS SM achieves a better one-to-one alignment with the observations than SMAPDA when evaluated by the in-situ data that used in its training process (USCRN and SCAN). However, the relative accuracy of GLASS SM and SMAPDA reverses when compared with the independent observations from OKMet, which implies the inclusion of physical constraints could be vital for a more consistent performance in SM estimate using the ML approach. From the analysis in anomalous

errors, we show similar intrinsic errors among the selected SM datasets in some cases, which is most likely driven by overlapping data sources. Referenced by the OKMet observations, an investigation on the realism of estimated SM spatial heterogeneity indicates all SM estimates, including the SMAPDA and GLASS SM, persistently underestimate the observed variances (~ 25% less) across the sites over the study period. While the ERA5-Land estimate shows larger and more comparable variances as observed, it does not accurately represent those SM values individually.

In addition to SM data, we showed that SMAPDA data reasonably represent ST and even surface heat fluxes when compared against the observations measured in ARM SGP sites. This suggests the suite of SMAPDA dataset is useful in characterizing land-atmosphere interactions. Moreover, it is also analyzed with respect to the response to a drought that occurred over the southeastern U.S. during the fall of 2016. As one of the key components contributing to the drought, the reduction in SM is usually accompanied by increased evaporation in the water-limited scenario, which may potentially amplify

and increase wildfire activity and stresses on agricultural production until new precipitation. We explored the relationships between SM and ET with a focus on Alabama quantitatively, utilizing concurrent GLEAMv4.1 data as the reference. Results indicate both datasets showed declined SM in summer and fall, with SMAPDA consistently displaying drier soil conditions compared to GLEAMv4.1. ET trends from both datasets were relatively close til June but diverged in summer, with SMAPDA estimating higher ET, exacerbating the drought conditions. Data also highlighted that model discrepancies, particularly in clay-

rich soils, suggest the need for refined treatments of hydraulic process in models for accurate SM estimates.

      A few uncertainties in our analysis are worth noting. For example, the evaluation result is most likely dependent on the data resolutions. Coarse resolution SM estimates such as ERA5-Land, GLEAMv4.1, and SMAP AM suffer from insufficient representativeness of subgrid SM variability, underscoring the necessity of high resolution to better characterize highly heterogeneous SM distributions. The mismatch of spatial resolutions among different data sources likely introduces uncertainty

in their intercomparison. More sophisticated upscale/interpolation algorithms may be employed to further mitigate the issue

(Crow et al. 2012; Gruber et al. 2020; Quiring et al. 2016), however as these methods introduce their own uncertainties, we opted not to use them for our evaluations. In addition, unresolved natural and anthropogenic processes such as surface and subsurface lateral flow (e.g, Yang et al. 2021), root water uptake and redistribution (e.g., Zeng 2001), dynamic groundwater water table and capillary rise (e.g., Miguez-Macho and Fan 2012), and irrigation (e.g., Yang et al. 2020) can potentially shift the SM estimates under various conditions. Along this line, the estimates in root-zone SM would be worth validated to further constrain the overall performance. While our SM dataset encompassing much of the eastern CONUS is restricted to a one-year period (2016), our results demonstrate a promising approach that can be applied to any local domain of interest with potentially longer analysis periods. This dataset could be used as lower boundary conditions to drive other meteorological model experiments that investigate the impact of land-atmosphere coupling on boundary layer properties and clouds. Lastly, there are many more ML algorithms, such as neural networks, random forests, and support vector machines, have been applied to enhance the spatial and temporal resolution of soil moisture datasets and further improve accuracy in data-sparse regions (e.g., O. and Orth 2021; Han et al. 2023; Lei et al. 2022; Zhang et al. 2023). However, such approaches lack the inherent physical constraints of a data-assimilation approach. Future studies may include more ML-based products in the assessment and discuss the impacts of physical constraint on estimated SM as suggested in this study.

**Competing interests**

The authors declare that they have no conflict of interest.

**Acknowledgments**

This research was supported by the Atmospheric Science Research (ASR) program as part of the DOE Office of Biological and Environmental Research. Pacific Northwest National Laboratory is operated by DOE by the Battelle Memorial Institute under contract DE-AC05-76RL01830. The modelling computation was performed on PNNL's Research Computing clusters. The authors would like to thank for all the technical supports through the LIS Github (https://github.com/NASA-LIS/LISF/discussions) and particularly the responsible research scientist David Mocko.

**Data availability**

The dataset generated and analyzed during the current study (Tai et al. 2025) is available on Zenodo at https://doi.org/10.5281/zenodo.14370563. The software package of the NASA Land Information System (LIS) can be downloaded through https://github.com/NASA-LIS/LISF (Kumar et al. 2006; Peters-Lidard et al. 2007). The enhanced SMAP Level 3 soil moisture product (SPL3SMP_E; O'Neill et al., 2020) is accessible at https://nsidc.org/data/spl3smp_e/versions/6. The NLDAS-2 data (Xia et al. 2012) is archived at https://disc.gsfc.nasa.gov/datasets/NLDAS_FORA0125_H_2.0/summary?keywords=NLDAS2. The Stage IV QPE product (Lin and Mitchell (2005)) is accessible at https://data.ucar.edu/dataset/ncep-emc-4km-gridded-data-grib-stage-iv-data. The

USCRN and SCAN data are acquired from the International Soil Moisture Network (ISMN; https://ismn.earth/en/, last access: 1 February 2024). The Oklahoma mesonet soil moisture observations (OKMet; McPherson et al. (2007)) were acquired from the ARM discovery website: https://adc.arm.gov//discovery/#/results/s::sgpokmsoilX1.c1 (ARM facility (1998)). All other SM datasets please refer to Table 3. The STAMP (Kyrouac et al., n.d., https://doi.org/10.5439/1238260) and ECOR (Gaustad, n.d., https://doi.org/10.5439/1097546) data were sourced from the Atmospheric Radiation Measurement (ARM) user facility.

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
