# Peer review of "A 1 km soil moisture data over eastern CONUS generated through assimilating SMAP data into the Noah-MP land surface model"

_Earth System Science Data, 2024_

## Author Response (AR1)

**Responses to Reviewer #1**

General comment:

This study assimilates the SMAP surface soil moisture product into the Noah-MP land surface model over the eastern continental U.S. for 2016 and evaluates its performance using multiple reference datasets, including model-driven products and in situ observations. The authors report strong overall performance of SMAP assimilation while noting that regions with higher errors tend to be associated clay soil types, highlighting directions for future improvements. The manuscript is well-structured and clearly written. However, there are two main weaknesses: 1) a lack of novelty in advancing soil moisture data assimilation research and 2) a lack of consideration on the spatial scale mismatch between model-derived products and in-situ observations. Please see my detailed comments below and I hope those will help enhance the manuscript.

We sincerely thank the reviewer for the thoughtful and constructive comments. The detailed feedback has been instrumental in improving the clarity, rigor, and overall quality of the manuscript. Our responses to each comment are provided below, directly following each bullet point. Reviewer's comments appear in regular black font and our responses appear in regular blue font.

Specific comments:

1. Introduction: The literature review primarily focuses on studies that assimilate SMAP into Noah-MP. However, it would be beneficial to broaden the discussion to include studies that 1) assimilate different soil moisture products and 2) employ DA with other land surface models, providing a more comprehensive overview of the benefits, limitations, and current challenges. And then clearly state the major contribution of this study to the soil moisture DA field. For instance, SMAP has a relatively coarse resolution, whereas other satellite-based soil moisture products such as Sentinel-based ones offer finer spatial scale. Why the authors choose to use SMAP for a 1-km assimilation set up? The introduction could be benefited from discussing how assimilating higher-resolution soil moisture products affects soil moisture estimation and how the assimilation of coarser SMAP data compares to these alternatives. Overall, the study's novelty and unique contributions relative to existing literature should be explicitly articulated.

Reply: Thank you for the constructive comments regarding the revisions for the introduction. The discussions have been extended accordingly on top of original texts to facilitate more inclusive descriptions. The revisions in response to the comments are detailed below:

- Include studies that 1) assimilate different soil moisture products and 2) employ DA with other land surface models:

Ans: We acknowledge the prior efforts in the field of satellite soil moisture assimilation had used a range of combinations in soil moisture retrievals and land surface models. We have included additional discussions to cover some of the great works in the paragraph from L73 to L80. A couple of sentences were also added in the paragraph from L84 to L89 for the justification of using Noah-MP as the core land surface model.

- Discussing how assimilating higher-resolution soil moisture products affects soil moisture estimation and how the assimilation of coarser SMAP data compares to these alternatives.

Ans: The existing SMAP-based 1-km soil moisture dataset (Lakashimi and Fang, 2023) could potentially provide finer-scale soil moisture info and may be incorporated into the assimilation process. However, as demonstrated in Fang et al. (2022), the spatial coverage and availability of the 1-km downscaled dataset is notably reduced compared to the 9-km dataset (please refer to Fig. 5 in Fang et al. (2022)).

Regarding the applications of Sentinel-1 based high-resolution (1-km) soil moisture datasets for data assimilation, there are multiple demonstrations such as those documented in Brocca et al. (2024), Filippucci et al. (2021), Foucras et al. (2020), Gao et al. (2017), and also the one jointly use SMAP and Sentinel-1 from Meyer et al. (2022). However, despite promising results, most of them were performed in a much smaller and local domain as opposed to those SMAP-relevant studies. This is primarily due to the contrasts in sensor characteristics between these two satellites (shown in the table below):

| Feature | SMAP | Sentinel-1 |
|---|---|---|
| Platform | NASA | ESA (Copernicus) |
| Sensor Type | Radiometer (passive) + Radar (active – failed in 2015) | Synthetic Aperture Radar (SAR, active) |
| Frequency | L-band (1.41 GHz) | C-band (5.4 GHz) |
| Resolution | ~36 km (radiometer), ~9 km (discontinued radar) | ~1 km (can be resampled to ~100 m) |
| Revisit Time | ~2-3 days | 6–12 days (depends on latitude and orbit) |

While Sentinel-1 has higher spatial resolution (~1 km) than SMAP (~9 km), it has relatively lower radiometry sensitivity and much longer revisit time (approximately 3 to 4 times longer), and notably it requires more complex preprocessing. That means, Sentinel-1 is most likely less sensitive to subtle differences in soil moisture content than SMAP and would be hardly capture some day-to day variability. Hence in general, SMAP is more suitable for regional/global scale applications, whereas Sentinel-1has more potential in local-scale monitoring. We summarize the discussions here in the revised manuscript from L98 to L109.

Reference:
Fang, B., Lakshmi, V., Cosh, M., Liu, P.-W., Bindlish, R., & Jackson, T. J (2022). A global 1-km downscaled SMAP soil moisture product based on thermal inertia theory. Vadose Zone Journal, 21, e20182. https://doi.org/10.1002/vzj2.20182

Lakshmi, V. and B. Fang. 2023. SMAP-Derived 1-km Downscaled Surface Soil Moisture Product, Version 1. Boulder, Colorado USA. NASA National Snow and Ice Data Center Distributed Active Archive Center. https://doi.org/10.5067/U8QZ2AXE5V7B.

Brocca, L., and Coauthors, 2024: Exploring the actual spatial resolution of 1 km satellite soil moisture products. Science of The Total Environment, 945, 174087, https://doi.org/10.1016/j.scitotenv.2024.174087.

Gao, Q., M. Zribi, M. J. Escorihuela, and N. Baghdadi, 2017: Synergetic Use of Sentinel-1 and Sentinel-2 Data for Soil Moisture Mapping at 100 m Resolution. Sensors, 17, 1966, https://doi.org/10.3390/s17091966.

Meyer, R., W. Zhang, S. J. Kragh, M. Andreasen, K. H. Jensen, R. Fensholt, S. Stisen, and M. C. Looms, 2022: Exploring the combined use of SMAP and Sentinel-1 data for downscaling soil moisture beyond the 1-km scale. Hydrology and Earth System Sciences, 26, 3337–3357, https://doi.org/10.5194/hess-26-3337-2022.

Filippucci, P., L. Brocca, R. Quast, L. Ciabatta, C. Saltalippi, W. Wagner, and A. Tarpanelli, 2022: High-resolution (1-km) satellite rainfall estimation from SM2RAIN applied to Sentinel-1: Po River basin as a case study. Hydrology and Earth System Sciences, 26, 2481–2497, https://doi.org/10.5194/hess-26-2481-2022.

Foucras, M., M. Zribi, C. Albergel, N. Baghdadi, J.-C. Calvet, and T. Pellarin, 2020: Estimating 500-m Resolution Soil Moisture Using Sentinel-1 and Optical Data Synergy. Water, 12, 866, https://doi.org/10.3390/w12030866.

2. Analysis domain and period: Since SMAP has been available from March 2015 onward, why was data assimilation performed for only a single year? A longer assimilation period could provide insights into the model's ability to represent interannual variability and extreme events such as drought. Since the authors did have a section discussing drought, I think it is not sufficient to discuss drought without referring to a long-term reference period. Please see my comment #12 in more detail.

   Reply: The study period was selected mainly due to the immediate modeling applications during the 2016 HI-SCALE field campaign as mentioned in Section 2. As mentioned in the summary and conclusion, the framework can be directly applied to any local region and produce long-term statistics over the area. In response to the comment #12, we use the approach suggested by the reviewer, which leverages the 5-year (2012 to 2016) long dataset from OL simulation in computing the reference climatology at each pixel over the AL (Alabama) subdomain. The corresponding analysis with respect to the 2016 drought event in Section 4.2.4 has also been revised based on the multi-year statistics. More details please refer to the reply for the comment #12 below.

3. L167-168: Is there a reference to support this claim, in particular reference that supports the evidence of Stage IV QPE being better than NLDAS-2 for the study period.

   Reply: Instead of citing a reference, we'd like to mention we carried out sensitivity experiments before applying to simulation for the entire study period. We examined the changes in soil moisture estimates due to the replacement in precipitation forcing data during the two intensive observation periods (IOPs) of HI-SCALE field campaign (Fast et al. 2018).

   The RMSEs were computed against several in-situ data collected by the Oklahoma Mesonet in a domain covering the ARM SGP site and adjacent area. The results in Fig. 1 and Fig. 2 demonstrate that soil moisture simulated by the OL simulation further improves after replacing the NLDAS-2 precipitation data as forcing. The comparison of instantaneous rain rate obtained from NLDAS-2 and Stage IV (ST4) precipitation at 00 UTC of August 30, 2016 (Fig. 3) also clearly shows that the Stage IV provides more heterogeneous precipitation distribution over the study domain.

[Figure]

Fig. 1 Root-mean-square error (RMSE) of soil moisture computed simulated by the open loop simulations using NLDAS-2 and Stage IV precipitation during the HI-SCALE IOP1 (April 24 − May 20, 2016).

[Figure]

Fig. 2 Similar to Fig. 1 but for HI-SCLAE IO2 (Aug. 28 − Sep. 23, 2016).

[Figure]

Fig. 3 Precipitation rate (unit: mm day$^{-1}$) maps extracted from NLDAS-2 (NLDAS2, left panel) and Stage IV (ST4, right panel) at 00 UTC on August 30, 2016.

Reference:

Fast, J. D., and Coauthors, 2018: Overview of the HI-SCALE Field Campaign: A New Perspective on Shallow Convective Clouds. Bull. Amer. Meteor. Soc., https://doi.org/10.1175/BAMS-D-18-0030.1.

4. Section 3.3: Some methodological details are unclear or missing: 1) Is there any spin-up process for the OL simulation, and does it reach equilibrium before applying the restart file for DA? 2) What is the reference period used for the monthly CDF matching? 3) Why use CDF-matching strategy? Does the CDF-matching strategy risk distorting real signals in SMAP data by adjusting them to fit the model-derived distribution? How do the CDF profiles compare to model-based CDF before and after CDF-matching? Some studies favor anomaly scaling over CDF matching to better preserve variability in satellite-based observations – additional analysis may be required to justify the choice of CDF matching.

Reply: Thanks for the comments regarding the DA strategies/methods. The answers to the bullets above are given below:

1)   Is there any spin-up process for the OL simulation, and does it reach equilibrium before applying the restart file for DA?

Ans: Yes, the OL simulation was performed with an over 4-year spin-up period from January 1, 2011, through March 31, 2015. The restart file generated at 2345 UTC on Mar. 31 was used for the DA simulation. The description of OL simulation is added in Section 3.3 from L245 to L248.

2)   What is the reference period used for the monthly CDF matching?

Ans: The reference period used for the monthly CDF matching is two years in total, ranging from Jan. 1, 2015, to Dec. 31, 2016. The relevant description has been added in the texts in L278.

3)   Why use CDF-matching strategy? Does the CDF-matching strategy risk distorting real signals in SMAP data by adjusting them to fit the model-derived distribution? How do the CDF profiles compare to model-based CDF before and after CDF-matching? Some studies favor anomaly scaling over CDF matching to better preserve variability in satellite-based observations – additional analysis may be required to justify the choice of CDF matching.

Ans: Anomaly matching involves assimilating the deviations from a long-term mean (e.g., SMAP anomaly vs. model anomaly), under the assumption that both datasets are centered, and variability is the key interest. It avoids altering the raw values of observations and thus preserves physical signals better. There are potential benefits when using the anomaly matching as opposed to CDF matching, including 1) better preserve actual observed event magnitudes and 2) avoid imposing model biases on observations. However, the implicit assumption of anomaly matching in neglecting the shifting between observational and model datasets is unlikely valid based on our analysis (annual mean values are notably different between raw SMAP data and SMAPDA). In general, the anomaly matching is more appropriate if the primary goal is to capture dynamics rather than absolute values. We also examined the sensitivity of CDF matching and found out the distortion wasn't an issue in our case. Hence, we decided to apply the CDF matching (Kumar et al. (2012) and Yang et al. (2020)) to the DA process in this study.

Reference:

Kumar, S. V., R. H. Reichle, K. W. Harrison, C. D. Peters-Lidard, S. Yatheendradas, and J. A. Santanello (2012), A comparison of methods for a priori bias correction in soil moisture data assimilation, Water Resour. Res., 48, W03515, doi:10.1029/2010WR010261.

Yang Y, Turner R, Carey-Smith T, Uddstrom M. A comparison of three model output statistics approaches for the bias correction of simulated soil moisture. Meteorol Appl. 2020; 27:e1970. https://doi.org/10.1002/met.1970

5. Evaluation: does the authors consider the correlation for anomalies by removing the monthly mean? Analyzing anomaly correlations could provide additional insight into performance, as seasonal cycles may dominate raw correlations. Besides, since monthly CDF matching is used, it might be also beneficial to compare OL against the reference datasets as correlation may be largely dominated by the seasonal variation from OL. Comparing OL vs. DA would help distinguish the improvement from DA.

Reply: Thanks for the suggestion on the inclusion of anomaly correlation coefficient (ACC) as an additional metric to help distinguish the improvement of DA from OL. Following the algorithm used in Kumar et al. (2009), we subtract the monthly-mean climatology of each dataset from the corresponding daily mean value. In this case, the anomalies represent the daily deviations from the mean seasonal cycle. Result shows robust improvement in anomaly correlation is found in most of the USCRN and SCAN sites (Fig. 4). The mean ACC across all sites from both observational networks increases approximately 0.05. The Section 3.4.1 and 4.2.1 have been significantly revised correspondingly.

[Figure]

Fig. 4 The difference of anomaly correlation coefficients (ACCs) computed between OL and SMAPDA experiments at site level by using USCRN (left) and SCAN (right) observations.

Reference:
Kumar, S. V., R. H. Reichle, R. D. Koster, W. T. Crow, and C. D. Peters-Lidard, 2009: Role of Subsurface Physics in the Assimilation of Surface Soil Moisture Observations. J. Hydrometeor., 10, 1534–1547, https://doi.org/10.1175/2009JHM1134.1.

6. L281-284 "...there are grids received zero updates, especially in the eastern part of the domain..." This needs clarification. Is this due to missing raw data, quality control procedures in SMAPDA, or another factor?

Reply: Great question. Based on the user guide (O'Neill et al., 2021), the SMAP Enhanced L3 Radiometer Global Daily 9-km EASE-Grid Soil Moisture (SPL3SMP_E) data are stored along with "retrieval quality flag" that tags quality of the soil moisture retrievals on each data sample. For example: 0: Retrieval successful; 1: Retrieval not attempted due to precipitation; 2: Retrieval not attempted due to dense vegetation; 3: Retrieval failed due to other reasons. Another "surface flag" provides information about the given surface conditions that may affect retrieval quality, which including presence of snow (ice), urban areas, frozen ground, and water bodies. The Land Information System (LIS) utilizes "retrieval quality flag" to determine if the SMAP soil moisture retrieval will be accepted for assimilation. Essentially, the LIS would assimilate observations where retrieval quality flag = 0 and reject observations with non-zero flags as those may denote issues such as precipitation, dense vegetation, or retrieval failure, ensuring only high-quality data are assimilated in the model.

The above descriptions are summarized and documented in the revised paragraph from L318 to L320.

Reference:
O'Neill, P. E., S. Chan, E. G. Njoku, T. Jackson, R. Bindlish, J. Chaubell, and A. Colliander. 2021. SMAP Enhanced L3 Radiometer Global and Polar Grid Daily 9 km EASE-Grid Soil Moisture, Version 5. Boulder, Colorado USA. NASA National Snow and Ice Data Center Distributed Active Archive Center. https://doi.org/10.5067/4DQ54OUIJ9DL.

L 285-291: It would be beneficial if the authors can elaborate the discussion on what might be causing the seasonal patterns of the increments. For instance, what processes might be captured by SMAP but not represented in the model? Does SMAP assimilation primarily correct model soil moisture errors originating from forcing uncertainties, or does it capture external influences such as irrigation, which are absent in the model? A deeper discussion of these factors with supported reference would strengthen the manuscript.

Reply: In general, the seasonal patterns of increments can be driven by:

1) Model deficiencies and missing physical processes:

These are often associated with the unrepresented surface processes. For example, while SMAP detects increased soil moisture due to irrigation, the Noah-MP LSM used in this study do not explicitly simulate irrigation (e.g., Felfelani et al. (2018) and Lawston et al. (2017)). In addition, practices like tilling or cover cropping affect surface moisture and are likely not captured by the model physics.

SMAP can also observe the effects of standing water or saturation from intense precipitation, which may not be realistically represented in models with simplified runoff schemes. In areas with a high water table, upward capillary rise can maintain higher surface soil moisture. If the model lacks a dynamic groundwater component, this influence is missed.

SMAP retrieval algorithms include corrections based on microwave interactions with vegetation. If the model uses outdated or static vegetation parameters (e.g., fixed LAI or land cover), that mismatch may show seasonally, especially during the green-up and senescence periods. Lastly, model could also under-resolve the influences of dynamic root as the root system is basically static over time.

2)  Errors in meteorological forcing:

There are most likely biases in radiation, temperature, and wind biases affect evapotranspiration estimates and thus soil moisture. Assimilation of SMAP data often compensates for these errors, especially after dry spells or in transition seasons (spring and fall).

Based on the above, we revised the paragraph from L329 to L336 in response to the comment.

Reference:
Felfelani, F., Pokhrel, Y., Guan, K., & Lawrence, D. M. (2018). Utilizing SMAP soil moisture data to constrain irrigation in the Community Land Model. Geophysical Research Letters, 45, 12,892–12,902. https://doi.org/10.1029/2018GL080870

Lawston, P. M., Santanello, J. A., Jr, & Kumar, S. V. (2017). Irrigation signals detected from SMAP soil moisture retrievals. Geophysical Research Letters, 44, 11,860–11,867. https://doi.org/10.1002/2017GL075733

7.  Section 4.2: The authors acknowledge to some extent that scale mismatch between SMAP (9 km resolution) and in situ observations could degrade evaluation metrics. Would it be possible to aggregate the relatively densely distributed OKMet or SCAN data to a resolution comparable to the model-based reference products and raw SMAP and reassess the metrics? For instance, how does the metrics change if evaluating at the ERA5-land or raw SMAP resolution? Addressing the scale mismatch explicitly would improve the fairness of the comparison and strengthen the conclusions.

Reply: The issue of scale compatibility in the intercomparison among in-situ (point) measurement, satellite footprint, and gridded land surface model output has been actively discussed in numerous literatures including those of Crow et al. (2012), Gruber et al. (2019), and Quiring et al. (2017). Basically, they demonstrated spatial resolution mismatch among different data sources indeed introduce uncertainty into evaluation/comparison. Nonetheless, as described in Crow et al. (2013), factors such as meteorological forcing, landcover patterns, topographic features, and soil textures could jointly contribute to the uncertainties when attempting to upscale from point-scale data to field, watershed, and regional scales (see Fig. 5). Since point (in-situ) observations do not evenly distribute in space, a simple average may also distort the result, thus weighting for each sample must be decided when upscaling (Fig. 6). However, the representativeness of each observation could vary from time to time as meteorology/precipitation is rather dynamic. Therefore, the upscaling procedure itself most likely would introduce another layer of uncertainty to the soil moisture field. Given the above, we decide to leave the evaluation result as is but emphasizing the importance and challenge in the upscale approach for assessment in Section 5 from L633 to L635.

[Figure]

Fig. 5 Dominant physical controls on soil moisture spatial variability as a function of scale. The gray shading of bars reflects the relative importance of each control at various scales with increasing intensity according to importance.

[Figure]

Fig. 6 Schematic for the configuration of the upscaling problem with respect to point measurements and a satellite footprint. The upscaling function F↑ is used to link a set of N = 4 point-scale ground observations to a spatial mean corresponding to the footprint scale of a satellite-based surface soil moisture retrieval. (Adopted from Figure 5 in Crow et al. (2012))

Reference:

Crow, W. T., A. A. Berg, M. H. Cosh, A. Loew, B. P. Mohanty, R. Panciera, P. deRosnay, D. Ryu, and J. P. Walker (2012), Upscaling sparse ground-based soil moisture observations for the validation of coarse-resolution satellite soil moisture products, Rev. Geophys., 50, RG2002, doi:10.1029/2011RG000372.

A. Gruber, G. De Lannoy, C. Albergel, A. Al-Yaari, L. Brocca, J.-C. Calvet, A. Colliander, M. Cosh, W. Crow, W. Dorigo, C. Draper, M. Hirschi, Y. Kerr, A. Konings, W. Lahoz, K. McColl, C.

Montzka, J. Muñoz-Sabater, J. Peng, R. Reichle, P. Richaume, C. Rüdiger, T. Scanlon, R. van der Schalie, J.-P. Wigneron, W. Wagner, Validation practices for satellite soil moisture retrievals: What are (the) errors?, Remote Sensing of Environment, Volume 244, 2020, 111806, ISSN 0034-4257, https://doi.org/10.1016/j.rse.2020.111806.

Quiring, S. M., T. W. Ford, J. K. Wang, A. Khong, E. Harris, T. Lindgren, D. W. Goldberg, and Z. Li, 2016: The North American Soil Moisture Database: Development and Applications. Bull. Amer. Meteor. Soc., 97, 1441–1459, https://doi.org/10.1175/BAMS-D-13-00263.1.

8. Figure 5 and 6: It is hard to visually distinguish the difference across the comparisons. Please consider keep the RMSE and Bias plot for SMAPDA vs. USCRN to illustrate the spatial pattern of the metrics while plot the RMSE and Bias differences for other four comparisons against that of SMAPDA vs. USCRN. This would help the audience better see locations where other datasets perform better or worse compared to SMAPDA. The same suggestions apply to the supplementary figures.

   Reply: Thanks for the suggestion on the visualization. We revised and reorganized figures as mentioned in the comments. Please see the changes of figures in the revised manuscript. Note that figure numbers are also changed, correspondingly.

9. Figure 9: The smaller STD in SMAP AM could be a consequence of the coarse spatial resolution. To ensure a fair comparison, it is better to scale OKMet to a comparable resolution before computing STD. Alternatively, the manuscript could discuss about the sensitivity of STD estimates at different spatial scales.

   Reply: Surface heterogeneity is one of the important metrics to infer the potential strength of land-atmosphere coupling, which motivated us to compute STD spatially. Indeed, different resolutions could affect how STD is computed. For instance, aggregation of soil moisture data tends to smooth out heterogeneity, potentially masking small-scale processes. On the other hand, scaling down from coarse to fine resolutions can introduce uncertainty and errors due to assumptions in spatial interpolation or downscaling algorithms. We aimed to keep the intercomparison in each of their own native grids, which provides insights into the raw performance of each estimate. Our analysis showed that model resolution may not be the primary factor that drives the overall STD. Even the coarser-resolution soil moisture analysis (e.g., ERA5-Land) can give you a larger STD than the fine-solution estimate (e.g., GLASS SM). Those can be attributed to differences in topography, soil texture, vegetation, land use, and meteorological forcing as well as model's physical processes. We've updated the texts in Section 5 (summary and discussion) from L633 to L635 to address this comment along with comment #7.

10. When assessing ST, LHF, and SHF, how does OL compare to SMAPDA? Since these variables are indirectly influenced by SMAPDA, it would be informative to first demonstrate the OL simulation performance to better understand the extent to which SMAP assimilation modifies these variables.

    Reply: We found the differences between OL and SMAPDA in these variables are rather marginal in absolute values. This is likely due to relatively small sample number within the confined area. Given that more general comparison between OL and SMAPDA has been covered in other parts of the revised manuscript, we decided to highlight the SMAPDA results as compared to the observations.

11. Section 4.2.4: A single year of data is insufficient to robustly assess drought conditions, as drought classification typically relies on long-term climatological baselines. Do the authors have a **longer-term OL simulation** with sufficient spin-up? Instead of relying solely on trends within a year, it would be more appropriate to rank the 2016 conditions relative to a multi-year climatology and analyze its deviation from historical conditions.

Reply: Thanks for the constructive suggestion. To obtain the reference climatology, we first use the 5-year (from 2012 through 2016) OL simulation results to provide the monthly climatological mean $(\overline{SM_m})$ and standard deviation $(\sigma_{SM,m})$ for each pixel in the AL subdomain:

$$\overline{SM_m} = \frac{1}{5} \sum_{y=1}^{5} SM_m$$

$$\sigma_{SM,m} = \sqrt{\frac{1}{5} \sum_{y=1}^{5} (SM_m - \overline{SM_m})^2}$$

Then, we computed the standardized soil moisture anomaly $(SMA_m)$ which helps the severity of drought at given time and space:

$$SMA_m = \frac{SM_m - \overline{SM_m}}{\sigma_{SM,m}}$$

The calculations are also applied to GLEAMv4.1 and the parallel comparison between SMAPDA and GLEAMv4.1 is given below. According to former studies such as Ontel et al. (2021), Jiménez-Donaire et al. (2020), and Tian et al. (2022), when SMA is between 0 and -1, the drought is considered mild; moderate drought if SMA lies between -1 and -2. Lastly, it is defined as severe drought when SMA below -2. It shows both SMAPDA and GLEAMv4.1 suggest moderate to severe drought conditions occurred in this region during the months of September, October, November (Fig. 7). Due to the contrast in sample size, SMAPDA demonstrates more spatial heterogeneity in SMA compared to GLEAMv4.1. This analysis reconfirms the drought period as defined in other relevant studies. The description for the additional analysis can be found in the paragraph from L541 to L549.

[Figure]

Fig. 7 Box and whisker plot for monthly standardized SM anomaly (SSA) computed for SMAPDA and GLEAMv4.1 data in 2016 using their own climatology (2012 - 2016) as references. Dashed lines denote the thresholds for the defined drought conditions.

**Responses to Reviewer #2**

This study assimilates SMAP (9 km) surface soil moisture estimates to generate 1 km soil moisture, fluxes (sensible, latent), and temperature estimates for a region (Central, SE) of CONUS that's of critical interest (high density of in situ networks) for drought monitoring. Other datasets (ERA5 [reanalysis], GLASS [ML], GLEAM [model], SMAP [remotely sensed]) are compared with the available in situ records for soil moisture or other land-atmosphere variables as available. The 1 km SMAPDA product shows relatively good performance, and reasons are suggest further improvement for soil moisture modeling. This study helps advance downscaling/modeling soil moisture with land surface models (specifically Noah-MP). However, as the first reviewer has mentioned, further discussion is required to help frame the significance of this experiment. The limited data generated also needs to be considered (1 year, 2016). How does comparing a 1 km SMAP DA-modeled soil moisture product against other soil moisture products (modeled or otherwise) for this 1 year contribute to either the LSM, soil moisture/climate community, or other stakeholders?

We sincerely thank the reviewer for the thoughtful and constructive comments. The detailed feedback has been instrumental in improving the clarity, rigor, and overall quality of the manuscript. Our responses to each comment are provided below, directly following each bullet point. Reviewer's comments appear in regular black font and our responses appear in regular blue font.

Major comments:

1. Why was SMAP 9 km used to downscale to a 1 km resolution? SMAP-based 1 km soil moisture datasets exist (i.e., the NSIDC-SMAP 1 km product by Fang et al., 2022) that is available for this study period and area. Sentinel-1 based soil moisture would also be good to consider/compare as well (Brocca et al., 2024).

Reply: Thanks for the comment. We acknowledge the existing SMAP-based 1 km soil moisture datasets could potentially provide finer-scale soil moisture info and may be incorporated into the assimilation process. However, as mentioned in Fang et al. (2022), the spatial coverage and availability of the 1-km downscaled dataset is notably reduced compared to the 9-km dataset (e.g., Fig. 1 as shown below).

Regarding the Sentinel-1 based high-resolution soil moisture datasets, there are multiple demonstrations such as those documented in Brocca et al. (2024), Filippucci et al. (2021), Foucras et al. (2020), Gao et al. (2017), and even the one jointly use SMAP and Sentinel-1 (Meyer et al. (2022)). However, despite promising results, most of them were generated and tested in a much smaller domain. This is primarily due to the contrasts in sensor characteristics between these two satellites (shown in the table below):

| Feature | SMAP | Sentinel-1 |
| --- | --- | --- |
| Platform | NASA | ESA (Copernicus) |
| Sensor Type | Radiometer (passive) + Radar (active – failed in 2015) | Synthetic Aperture Radar (SAR, active) |
| Frequency | L-band (1.41 GHz) | C-band (5.4 GHz) |
| Resolution | ~36 km (radiometer), ~9 km (discontinued radar) | ~1 km (can be resampled to ~100 m) |
| Revisit Time | ~2-3 days | 6–12 days (depends on latitude and orbit) |

While Sentinel-1 has higher spatial resolution (~1 km) than SMAP, it has relatively lower radiometry sensitivity, much longer revisit time (approximately 3 to 4 times longer), and requires more complex preprocessing. This means Sentinel-1 is most likely less sensitive to subtle differences in soil moisture content and have limited ability to capture day-to day variability. In this case, SMAP is more suitable for regional-to-global scale applications, whereas Sentinel-1has more potential in local-scale monitoring.

Given the above, we did not attempt to assimilate any of the downscaled (higher spatial resolution) datasets. To acknowledge that those emerging data/products may be potentially used in the future, we've summarized discussions here and added into the introduction from L98 to L109.

[Figure]

Fig. 1 Comparison of 1-km and 9-km SMAP soil moisture composite maps for the periods of Jun 1 – 8, 2018 and Jun 9 – 16, 2018. (adopted from Fig. 5 in Fang et al. (2022))

Reference:

Fang, B., Lakshmi, V., Cosh, M., Liu, P.-W., Bindlish, R., & Jackson, T. J (2022). A global 1-km downscaled SMAP soil moisture product based on thermal inertia theory. Vadose Zone Journal, 21, e20182. https://doi.org/10.1002/vzj2.20182

Lakshmi, V. and B. Fang. 2023. SMAP-Derived 1-km Downscaled Surface Soil Moisture Product, Version 1. Boulder, Colorado USA. NASA National Snow and Ice Data Center Distributed Active Archive Center. https://doi.org/10.5067/U8QZ2AXE5V7B.

Brocca, L., and Coauthors, 2024: Exploring the actual spatial resolution of 1 km satellite soil moisture products. Science of The Total Environment, 945, 174087, https://doi.org/10.1016/j.scitotenv.2024.174087.

Gao, Q., M. Zribi, M. J. Escorihuela, and N. Baghdadi, 2017: Synergetic Use of Sentinel-1 and Sentinel-2 Data for Soil Moisture Mapping at 100 m Resolution. Sensors, 17, 1966, https://doi.org/10.3390/s17091966.

Meyer, R., W. Zhang, S. J. Kragh, M. Andreasen, K. H. Jensen, R. Fensholt, S. Stisen, and M. C. Looms, 2022: Exploring the combined use of SMAP and Sentinel-1 data for downscaling soil moisture beyond the 1 km scale. Hydrology and Earth System Sciences, 26, 3337–3357, https://doi.org/10.5194/hess-26-3337-2022.

Filippucci, P., L. Brocca, R. Quast, L. Ciabatta, C. Saltalippi, W. Wagner, and A. Tarpanelli, 2022: High-resolution (1-km) satellite rainfall estimation from SM2RAIN applied to Sentinel-1: Po River basin as a case study. Hydrology and Earth System Sciences, 26, 2481–2497, https://doi.org/10.5194/hess-26-2481-2022.

Foucras, M., M. Zribi, C. Albergel, N. Baghdadi, J.-C. Calvet, and T. Pellarin, 2020: Estimating 500-m Resolution Soil Moisture Using Sentinel-1 and Optical Data Synergy. Water, 12, 866, https://doi.org/10.3390/w12030866.

2. Why is only SMAP AM used for the assimilation? The afternoon product is mentioned, but I did not see any further justification for excluding it. Also, note that the SMAP L3 product is not exactly daily due to missing coverage. Was any gap-filling used prior to the data assimilation?

Reply: We would like to clarify that the exclusion of SMAP PM overpass was only employed when analyzing the intercomparison among different soil moisture data/products. The assimilation process applies all valid overpasses, both AM and PM.

There was no gap-filling applied prior to the data assimilation. Based on the time label in the SMAP data, the assimilation was performed at those pixels with valid SMAP data coverage in an hourly frequency.

3. The resolution of each respective product appears to be an important point for explaining the pattern in performance (e.g., GLASS & SMAPDA vs. SPL3SMP_E/ERA5-Land/GLEAM). Why didn't the authors attempt a higher resolution grid (e.g., 500 m) for the forcing? Given the Rouf et al. (2021) paper investigated up to that resolution with Noah-MP, such a study may be more impactful than the current one presented. Their study also cite increasing accuracy with increased forcing resolution.

Reply: Thank you for pointing out the potential sensitivities due to the model resolutions and forcing data.

Rouf et al. (2021) used physically based downscaling method to obtain 500-m forcing data from 12.5 km resolution NLDAS-2 atmospheric variables (air temperature, pressure and humidity; longwave

and shortwave radiation; wind speed and wind direction) and demonstrated positive impacts on soil moisture results. However, the downscaled forcing remains model-based and did not include fine-scale precipitation (remains in 12.5 km grid spacing). The coarse resolution precipitation could introduce large uncertainty in simulating soil states as precipitation amount at each grid point determines how much water may penetrate to soil layers. Thus, as opposed to the approach taken in Rouf et al. (2021), we replaced the precipitation forcing in the NLDAS-2 by Stage IV (grid spacing of 4 km), a radar-observation-constrained precipitation data valid for the CONUS region.

To justify the use of Stage IV precipitation data, we examined the changes in soil moisture estimates due to the replacement in precipitation forcing data during the two intensive observation periods of HI-SCALE field campaign (Fast et al. 2018). The RMSEs were computed against several in-situ data sets collected by the Oklahoma Mesonet in a domain covering the ARM SGP site and adjacent area. The results in Fig. 2 and Fig. 3 demonstrate that soil moisture simulated by the OL simulation further improves after replacing the NLDAS-2 precipitation data in the forcing data with the Stage IV precipitation data. The comparison of instantaneous rain rate obtained from NLDAS-2 and Stage IV (ST4) precipitation at 00 UTC of August 30, 2016 (Fig. 4) also clearly shows that the Stage IV provides more heterogeneous precipitation distribution over the study domain.

[Figure]

Fig. 2 Root-mean-square error (RMSE) of soil moisture computed simulated by the open loop simulations using NLDAS-2 and Stage IV precipitation during the HI-SCALE IOP1 (April 24 - May 20, 2016).

[Figure]

Fig. 3 Similar to Fig. 1 but for HI-SCLAE IO2 (Aug. 28 - Sep. 23, 2016).

Fig. 4 Precipitation rate (unit: mm day$^{-1}$) maps extracted from NLDAS-2 (NLDAS2, left panel) and Stage IV (ST4, right panel) at 00 UTC on August 30, 2016.

Reference:

Fast, J. D., and Coauthors, 2018: Overview of the HI-SCALE Field Campaign: A New Perspective on Shallow Convective Clouds. Bull. Amer. Meteor. Soc., https://doi.org/10.1175/BAMS-D-18-0030.1.

4. Why was only the upper soil moisture compared? I understand SMAP is the limiting factor here, but I'd argue the benefit of a (DA) model is to investigate the variables (e.g., root zone soil moisture) not directly sensed by the satellite. Given the comparison with ERA5-Land, soil moisture at lower depths would be helpful for comparison/validation as well.

Reply: We acknowledge that evaluation/comparison in the context of root-zone soil moisture would potentially provide more insights into how model/assimilation acted to represent soil moisture in deeper soil layers. Nevertheless, as you noted, SMAP and GLASS SM provide soil moisture estimates only for the surface layer, and other products and in-situ observations use different depths or layer thicknesses to report soil moisture. In-situ observations likely report a point at a vertical level (e.g., 25 cm), while the interpretation of model-based products is rather unclear (point or layer-average?). The table here summarizes the depths/layers of soil column used in the models and in-situ measurements:

| Dataset | Depths/layers (cm) | Notes |
| --- | --- | --- |
| SMAPDA | 10, 30, 60, 100 | 4-layer model output |
| ERA5-Land | 0–7, 7–28, 28–100, 100–289 | 4-layer model output |
| USCRN | 5, 10, 20, 50, 100 | In-situ sensors |
| SCAN | 5, 10, 20, 50, 100 | In-situ; may vary by site |
| OKMet | 5, 25, 60 | In-situ; fewer layers |

Therfore, unless different datasets report soil moisture at similar depths or report vertically integrated water amount (which seems complicated for in-situ measurements), we feel it is difficult to make a fair comparison and draw a firm conclusion for deeper soil layers. Having said that, other ongoing work

compares soil moisture simulated in Large-Eddy simulations using SMAPDA as the soil initial condition to in-situ measurements at ARM SGP site to assess SMAPDA's realism in the deeper soil layers. Also, we discuss the potential of this further analysis in the Section 5 (Summary and discussion) from L638 to L639."

5. An open-loop simulation would be important for a base comparison. How do you justify not running OL simulations and comparing them against the DA outputs? Furthermore, more details on the modeling set-up could be provided. How much spin-up time was used? Although the purpose for analyzing data from 2016 is given (major drought year), either more work or reasoning needs to be done for limiting the study period to one year. E.g., how do you know your results are representative of the datasets themselves, and not just valid for times of drought?

Reply: Thank you for commenting on the absence of open-loop (OL) simulation in the analysis. To address the comment, we've extensively updated the result section with additional results obtained from OL simulation. Meanwhile, the Section 3.3 was also revised to include more descriptions about the model setup for the OL simulation.

In this revision, the 5-year (2012 to 2016) outputs from OL simulation are also leveraged in the computation of climatological references in the discussion of the 2016 drought in southeastern U.S. More details please refer to the paragraph between L541 and L549.

Minor comments:

6. The number of stations should be explicitly described, and/or mention the OK (the state) in Figure 1 when describing the ARM SGP sites. The white circles were a little difficult to identify at first. The number of stations for validation would also help contextualize Figure 4.

Reply: The number of observational sites for Oklahoma Mesonet and ARM SGP are added for clarification. The sentence in L206 is revised accordingly as: "The OKMet (120 sites) and ARM SGP (6 sites) observations are adopted…"

7. Line 45 - is (O (10 km)) a typo?

Reply: No, it represents it is in the order of 10 kilometers.

8. Line 320, Figure 4 - GLASS is stated to have more off-diagonal terms, but the concentration and shape of off-diagonal terms in the SMAPDA plot make that statement appear moot. The total RMSE for GLASS is also slightly lower than the SMAPDA product.

Reply: Good catch. We decide to drop this statement.

9. Line 482 - Sites 6, 8, 14 also show high bias during some months but are more sandy soils. What do you attribute the cause to be? Could it be a question of sampling? (e.g., less clay sampled available, whereas more sandy time series exist)

Reply: Although the sites 6, 8, and 14 also show relatively larger differences between models and observations compared to other sites with sandy soil, the observed soil moisture does not exceed 0.5

mm mm⁻³ at those sites. Whereas maximum soil moisture in the three clay-soil sites can be even larger than 0.6 mm mm⁻³.

To understand if this conclusion is statistically robust, we examined the results for all the sites (USCRN and SCAN) located in the study domain. Fig. 5 shows the RMSE of SMAPDA soil moisture computed in a function of soil types. Although the sample size varies among soil types, the soil moisture error is generally larger at sites with clay soil than other soil types. An extended analysis focusing on its dependency on landcover types (Fig. 6) further shows the differences in error statistics in association with landcover types are less distinct. This suggests that our hypothetical conclusion is likely true despite other factors could potentially contribute to the errors. The description of extended analysis is included in the revised manuscript from L569 to L571.

[Figure]

Fig. 5 SMAPDA soil moisture RMSE computed for all SCAN and USCRN sites as a function of soil texture. The number of sampling sites for each soil texture type is given in the parenthesis.

[Figure]

Fig. 6 Similar to Fig. 5 but for landcover types.

10. Line 444 - The soil moisture bias doesn't appear to be that great compared to the in-situ data. And LHF is only marginally larger for SMAPDA than the observed during the summer. The differences in the summer SHFs also show that they are not within error/standard deviations. Are there other causes you could attribute this discrepancy to?

Reply: According to the equation of SHF, the magnitude of SHF is primarily driven by the temperature gradient between the ground surface and the air:

$$SHF = \rho C_p \frac{T_s - T_a}{r_a}$$

Where $T_s$ is the surface temperature (K), $T_a$ represents air temperature (K). $\rho$ is air density (J kg$^{-1}$ K$^{-1}$), whereas $C_p$ stands for specific heat capacity of air (kg m$^{-3}$). $r_a$ denotes aerodynamic resistance (s m$^{-1}$), which is affected by wind speed, surface roughness, atmospheric stability. Note that the SHF may be

changed drastically when the temperature gradient shifts direction or magnitude. As also noted by Xie et al. (2012, 2014), uncertainties in meteorological forcing (NLDAS-2) in terms of air temperature, wind, and stability can potentially modulate the difference in SHF between SMAPDA and local observations during the summer months, when local land-atmosphere interactions become more important than in other seasons. For example, the NLDAS-2 forcing is based on the North American Regional Reanalysis on the 32-km grid, and downscaled to 1/8° grid, or ~ 12 km grid spacing; this grid spacing is not enough to capture the local-scale variability of near-surface meteorological variables happening at point-scale observations.

Another uncertainty is from the measurement, for which we only used the flux measurement from the eddy covariance (EC) systems. It was noted that there can be large differences between the fluxes from EC and those estimated by the Bowen-Ratio energy balance approach. In this study, we focused on eddy covariance measurement that is widely used in other national/international network (e.g., FLUXNET). Lastly, Noah-MP provides an extensive range of model configurations, e.g., bottom boundary condition for the soil layers (with or without ground water), soil hydrology, bare-ground evaporation resistance, canopy radiative transfer, etc., which will carry the impact of soil moisture assimilation differently to the surface flux."

---

## Author Response (AR2)

We sincerely thank the two reviewers for the second review. Our responses to each comment from both reviewers are given below, directly following each bullet point. Reviewer's comments appear in black font and our responses appear in blue font.

**Responses to Reviewer #1**

The authors did a good job in addressing my previous comments and I don't have further comments for this manuscript besides the following minor suggestion:

L103-104: please double check if the reference is Feng et al. (2020) or Feng et al. (2022).

Reply: Thanks for the catch! That seems to be a glitch in the Zotero. We managed to resolve and thus it has been corrected.

**Responses to Reviewer #2**

The revised study and authors made good efforts to address past concerns and specific questions, especially by adding the open loop simulations. However, some minor comments remain below. This study may be helpful for those requiring decision making in this study region of CONUS and helps contextualize the different soil moisture products available for the upper layer. The comparison between outputs based on a physics-based model and a ML-based model is helpful for understanding the drawbacks/benefits of either.

Minor comments:

1. It would be good to show the figures for justifying the NCEP Stage IV precipitation product as shared in the author's responses for Section 3.2.3 at least in the supplementary document. This is to help readers who may not check the reviewer's comments otherwise.

   Reply: The figure showcasing the contrast between the NCEP Stage IV and NLDAS-2 precipitation rates are now valid in the supplementary document as the new Figure S1. Additional description of this figure is added in L203.

2. Figure 8 (and Figures S5, 10) may be further improved with the addition of the OL timeseries, to contextualize how much improvement the SMAP DA was compared to it.

   Reply: The three figures have been updated with the addition of the OL timeseries, correspondingly.

3. A statement on why only GLEAMv4.1 is used to compare the drought event in 2016 against SMAP DA should be made (and not including ERA5-L, GLASS, and SMAP AM, which were analyzed in the previous sections).

Reply: A statement explains why the GLEAMv4.1 was chosen as the primary benchmark in the specific comparisons can be found in L532 – L533.

4. The RMSE for land cover types in the response (Figure 6) is recommended to also be included in the supplementary section.

Reply: Thanks for the suggestion. The figure illustrating landcover-dependent SM RMSEs now appears in the supplementary material document as Figure S9. The relevant discussions in section 4.2.4 have also been revised accordingly.